# MindAgent: Emergent Gaming Interaction

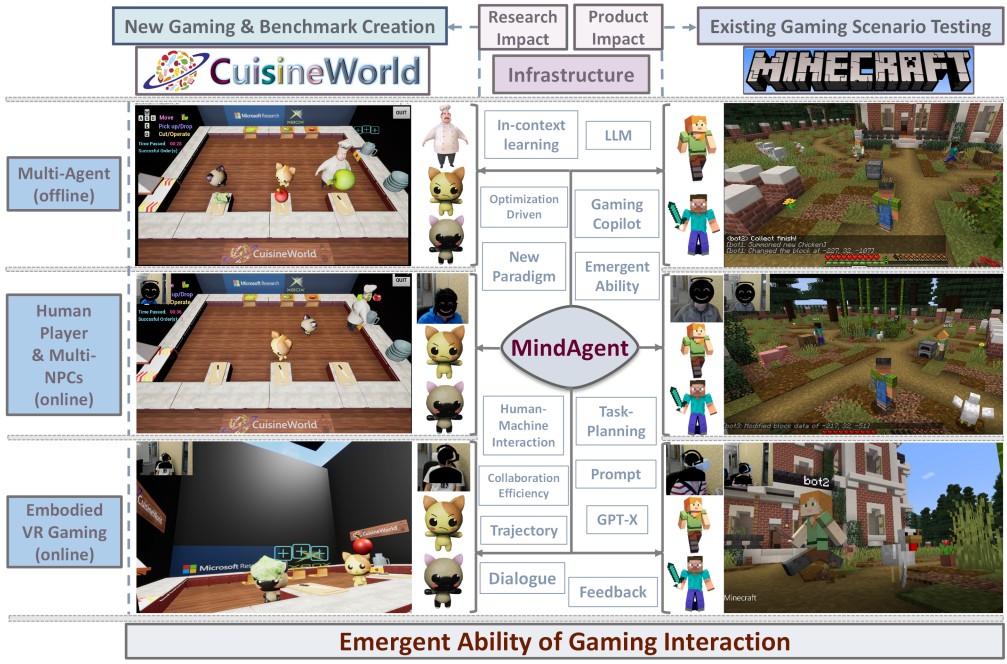

Figure 1: The MindAgent system for gaming interactions. MindAgent enables complex task planning in a multi-agent system and provides a human-AI collaboration infrastructure across various domains.

## Abstract

Large Language Models (LLMs) can perform complex scheduling in a multi-agent system and can coordinate agents to complete sophisticated tasks that require extensive collaboration. However, despite the introduction of numerous gaming frameworks, the community lacks adequate benchmarks that support the implementation of a general multi-agent infrastructure encompassing collaboration between LLMs and human-NPCs. We propose a novel infrastructure—**MindAgent**—for evaluating planning and coordination capabilities in the context of gaming interaction. In particular, our infrastructure leverages an existing gaming framework to (i) require understanding of the coordinator for a multi-agent system, (ii) collaborate with human players via instructions, and (iii) enable in-context learning based on few-shot prompting with feedback. Furthermore, we introduce **CuisineWorld**, a new gaming scenario and its related benchmark that supervises multiple agents playing the game simultaneously and measures multi-agent collaboration efficiency. We have conducted comprehensive evaluations with a new auto-metric *collaboration score* **CoS** for assessing the collaboration efficiency. Finally, MindAgent can be deployed in real-world gaming scenarios in a customized VR version of CuisineWorld and adapted in the broader "Minecraft" gaming domain as showed in Figure 1. Our work involving LLMs within our new infrastructure for general-purpose scheduling and coordination can elucidate how such skills may be obtained by learning from large language corpora.

## 1 INTRODUCTION

Large language Models (LLMs) have been driving the effort to develop general intelligent machines (Bubeck et al., 2023; Mirchandani et al., 2023). Although they are trained using large text corpora, their superior problem-solving capacity is not limited to canonical language processing domains. LLMs can potentially tackle complex tasks that were previously presumed exclusive to human experts or domain-specific algorithms. Recent research has shown the possibility of using LLMs to generate complex plans for robots and game AI (Liang et al., 2022; Wang et al., 2023b;a; Yao et al., 2023; Huang et al., 2023), marking an important milestone for LLMs as general-purpose intelligent agents. In this paper, we investigate the planning capacity of LLMs in the context of multi-agent systems (Stone & Veloso, 2000). Compared to planning for a single agent, which has been studied extensively (Wang et al., 2023b;a), multi-agent planning imposes much higher problem-solving complexity due to an action space that grows exponentially with respect to the number of agents. The planner must simultaneously control multiple agents, avoid possible conflicts, and coordinate agents into achieving a shared goal that requires potentially sophisticated collaboration. To understand to what extent LLMs can acquire multi-agent planning skills, we first develop a new benchmark, **CuisineWorld**, which is illustrated in Figure 1.

To incorporate agent AIs into video games, we design **MindAgent**, an infrastructure inspired by multi-agent task allocation optimization theories, to facilitate the multi-agent planning capabilities of LLMs. Our infrastructure enables LLMs to perform complex coordination and scheduling of multiple agents in order to achieve task completion. We conduct comprehensive evaluations with recently introduced LLMs, including GPT-4, Claude, and LLaMA, playing our CuisineWorld game within our MindAgent interactive multi-agent planning framework, leading to the following key observations: 1) **Zero shot multi-agent planning**: Powerful pretrained LLMs like GPT-4 are capable of scheduling multiple agents (ranging from 2 to 4) to complete dishes, even by collaborating with human players, by merely reading game instructions and recipes; 2) **Planning with advanced prompting**: We can significantly boost multi-agent planning performance by leveraging an emergent *in-context learning* ability (Brown et al., 2020; Wei et al., 2021) by adding only a few expert demonstrations (from different games) to the prompt, explaining the rationale of certain actions as in Chain-of-Thought prompting (Wei et al., 2022), and providing on-the-fly feedback to the LLMs during planning; and 3) **Generalization**: LLMs can potentially be generalist multi-agent planners as they are able to generalize in order to coordinate a growing number of agents and perform well in new game domains such as Minecraft.

The main contributions of our work are as follows:

- We develop a new gaming scenario and related benchmark based on a multi-agent virtual kitchen environment, CuisineWorld. It adopts a minimal text-based game format and supports planning tasks with various structures and challenges, making it an ideal test bed for the emergent multi-agent planning (i.e., scheduling and coordination) capacity of LLMs.

- We introduce MindAgent, an infrastructure for interactive multi-agent planning with LLMs. which demonstrates the in-context learning of the multi-agent planning capacity of LLMs and offers several prompting techniques to facilitate their planning ability, including providing few-shot demonstrations, planning rationals, and environmental feedback.

- We conduct extensive evaluations of our benchmark with multiple LLMs and prompting settings. Our experimental results validate its potential in helping develop generalist multi-agent planners.

- We deploy MindAgent in real-world gaming scenarios and demonstrate its ability to power human-AI interactions.

Compared to canonical domain-specific automated planning systems, although multi-agent planning with LLMs is more likely to be bottlenecked by high computational cost, context length limitations, non-optimal plans, *etc.*, it can potentially improve planning performed by *in-context learning* from data without fine-tuning, seamlessly adapt to new planning problems across different domains, and offer a more flexible interface to human collaborators. Ultimately, our investigation into the leveraging of LLMs for general-purpose scheduling and coordination can elucidate how such skills may be acquired by learning from large text corpora, and is potentially instrumental to the future development of more effective LLM-based planners.

## 2 RELATED WORK

**Multi-Agent Coordination.** The field of multi-agent collaboration boasts a comprehensive body of literature. Traditionally, such collaborations have been modeled using the MDP/POMDP frameworks (Lowe et al., 2017; Rashid et al., 2020; Jain et al., 2019; Wu et al., 2021; Gao et al., 2023). However, there has been a recent shift towards using LLMs for these collaborations. For instance, Zhang et al. (2023b) delved into how LLMs might communicate and cooperate in a watch-and-help (WAH) task. Meanwhile, Zhang et al. (2023a) investigated a two-agent collaboration game inspired by the simpler dynamics of the two-agent Overcooked-style game. Notably, their research mainly concentrated on the task success rate, with most studies typically anchored to a single task objective. By contrast, we emphasize the importance of collaboration efficiency in scenarios encompassing multiple task objectives. Further, our research uniquely focuses on evaluating the collaborative efficiency of two or more agents. Additionally, while other works such as that of Park et al. (2023); Wu et al. (2021) simulate each agent individually, we employ a centralized system. This not only significantly reduces the number of API calls but also reduces context length, making it more appropriate for use in gaming applications.

**Planning With LLMs.** A number of works leverage LLMs to perform task planning (Huang et al., 2022a; Wang et al., 2023a; Yao et al., 2023; Li et al., 2023), specifically the LLMs' WWW-scale domain knowledge and emergent zero-shot planning abilities to perform complex task planning and reasoning. Recent robotics research also leverages LLMs to perform task planning (Ahn et al., 2022; Huang et al., 2022b; Liang et al., 2022) by decomposing natural language instruction into a sequence of subtasks, either in the natural language form or in Python code , then using a low-level controller to execute these subtasks. Additionally, Huang et al. (2022b), Liang et al. (2022), and Wang et al. (2023b) also incorporate environmental feedback to improve task performance.

**Benchmarks Using Games.** Numerous games have been developed to study task planning (Baker et al., 2022; Carroll et al., 2019; Bakhtin et al., 2022), yet only a handful delve into multi-agent collaborations. Even within this limited subset, the focus predominantly remains on two-agent interactions where responsibilities are unevenly distributed between the agents (Wan et al., 2022; Puig et al., 2020)—it is common for one player to assume a dominant role while the other provides support. By contrast, our work assumes the equal apportion of responsibilities across agents, and we expand our investigation to encompass collaborations involving more than two agents, even including human players. While some previous studies have ventured into multi-task settings, none has delved into scenarios where agents must compete for resources to complete multiple distinct tasks with varied levels of difficulty within a single episode. Additionally, our work differs from that of Carroll et al. (2019) in that our game settings feature a diverse array of tools and task objectives, thereby generating an exponentially larger task space. A comparison between our work and other related studies can be found in the Appendix E.1.

## 3 THE CUISINEWORLD GAME

We introduce CuisineWorld as a novel and flexible game for multi-agent scheduling and coordination in a *virtual kitchen* environment. In this game, a multi-agent system must supervise multiple agents and coordinate them, with the goal of completing as many dish orders as possible. The game is equipped with a textual interface since our focus is on evaluating LLM-based planning agents. Our modularized design separates tasks and game engines, allowing inclusion of more tasks (dish types) and domains ("kitchen" implementation via text-based engine, Unity, Minecraft, *etc.*).

**Tasks and Reward.** A task in CuisineWorld is a dish order, ranging from the most basic `tunaSashimi`, which can be made by simply chopping raw tuna meat, to sophisticated dishes like `porkPasta` requiring various cooking tools. In a game episode with a maximum of $T$ steps, in every *task interval* $\tau_{\text{int}}$, a new task or dish order will be added to the active task list. A task will be regarded *completed* and be removed from the active task list when a suitable dish has been placed on the serving table. On the contrary, a task will be deemed to have *failed* and be removed from the list after its *lifetime* $\tau_{\text{lft}}$, which depends on the complexity of the dish, is exceeded. Along with the tasks, the game provides rewards and penalties or feedback on certain occasions, *e.g.* when a task is just completed, when infeasible commands are dispatched, *etc*. We support five different actions 1) goto 2) get 3) put 4) activate 5) noop. The state space contains descriptions of the environment and agents. Due to space limitations, we refer the reader to additional details in Appendix D.

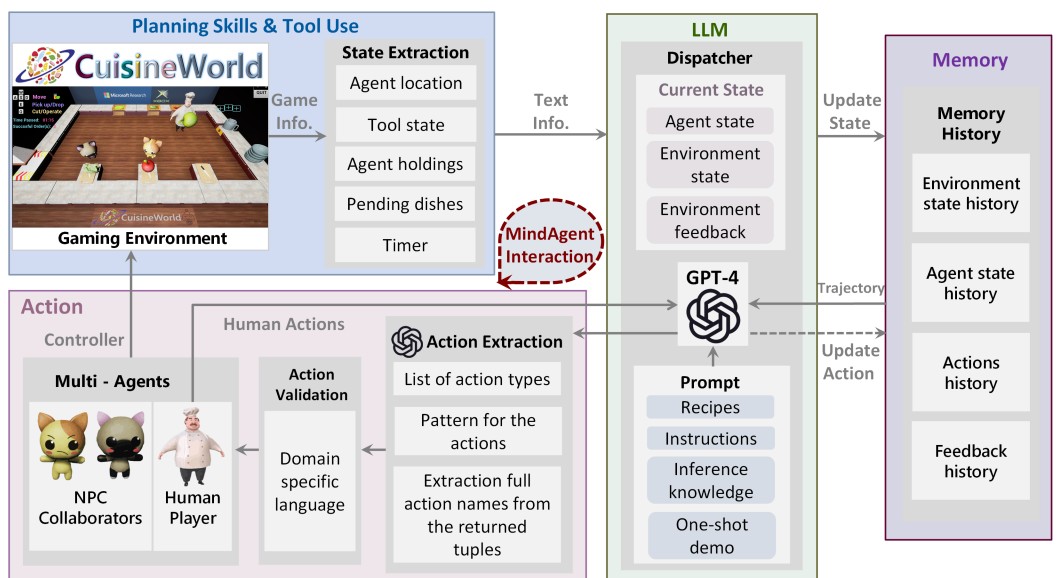

Figure 2: The MindAgent Infrastructure. **Planning Skill and Tool Use:** The game environment requires diverse planning skills and tool use to complete tasks. It generates relevant game information and converts the game data into a structured text format that the LLMs can process. **LLM:** The main workhorse of our infrastructure makes decisions, thus serving as a dispatcher for the multi-agent system. **Memory History:** A storage utility for relevant information. **Action Module:** Extracts actions from text inputs and convertd them into domain-specific language and validates DSLs so that they cause no errors during execution.

**Collaboration Score (CoS).** We need to evaluate to what extent the dispatcher (played by an LLM) can coordinate multiple agents to complete dish orders across a variety of scenarios. We are particularly interested in the question: Can the dispatcher continue to coordinate the agents into efficient collaborations with decreasing $\tau_{\text{int}}$; *i.e.*, as more dish orders are flooding in? Our hypothesis is that an ideal dispatcher should be capable of coordinating the agents until there are way more tasks than the system can handle. Therefore, we introduce a *collaboration score* (**CoS**), defined as

$$\text{CoS} = \frac{1}{M} \sum_{i=1}^{M} \frac{\text{Number of completed tasks} \left[\tau_{\text{int},(i)}\right]}{\text{Number of completed tasks} \left[\tau_{\text{int},(i)}\right] + \text{Number of failed tasks} \left[\tau_{\text{int},(i)}\right]} \quad (1)$$

where $M$ is the total number of $\tau_{\text{int}}$ intervals evaluated. Effectively, CoS is the average task completion rate across different $\tau_{\text{int}}$ conditions. In our default setting, we use $M = 5$. While the actual values of $\tau_{\text{int}}$ depend on the game level, we ensure that they span a wide range of difficulties including both relaxed and intense scenarios.

In summary, CuisineWorld is a game that emulates a virtual kitchen in which several robotic agents are commanded to use various cooking tools and ingredients to prepare as many dish orders as possible in a limited period of time. To necessitate collaboration, new orders will keep flooding in while the existing ones should be completed before their expiration times. Therefore, LLMs must properly coordinate the agents to maximize overall productivity. CuisineWorld offers game levels with a wide range of planning difficulty: dishes with different complexity (number of ingredients and tools involved), number of agents, order frequency and lifetime, etc., making it a useful test bed for LLM-based multi-agent planning.

## 4 THE MINDAGENT GAMING AI INFRASTRUCTURE

Our first foray into the challenging CuisineWorld benchmark is an interactive multi-agent planning framework with LLMs. It facilitates in-context learning and adopts a minimalist design for the purposes of demonstrating the emergent scheduling and coordination capacity while also bringing in exploratory prompting techniques that facilitate better planning and inform future approaches in this domain. Our MindAgent infrastructure comprises prompt, current state, and memory components, as shown in Figure 2 with details illustrated as follows:

**Prompt** incorporates four distinct sub-components: recipes, general instructions, inference knowledge, and a one-shot demo. **Recipes** outline hierarchical procedures for preparing various dishes at a given level. They specify the ingredients necessary for each intermediate or final product, the appropriate tools, and the expected post-cooking outcome. **Instructions** detail the foundational

rules of CuisineWorld, delineating the array of actions agents can undertake within the game and enumerating the characteristics of every tool available in the current kitchen scenario. Moreover, they inform agents about the base ingredients retrievable from storage, as well as all potential intermediate products they can procure. Agents are also explicitly advised to remain cautious about feedback from the environment. **Inference Knowledge** encapsulates insights and helpful hints for the agent, which when utilized appropriately can guide agents to sidestep potential errors and improve their collaborative efficiency. **One-shot Demo** presents a step-by-step demonstration of the preparation of a distinct dish, different from other dishes at the current level, spanning several time steps, each of which is incorporated as part of the prompt. The demonstration illustrates the major procedures for cooking a dish in CuisineWorld, including obtaining ingredients, putting ingredients into different tools, transporting intermediate ingredients, and delivering the final dish to the serving table. More details of prompt please find in Appendix A.

**Current State** provides a snapshot of the prevailing observations from the environment. It encompasses information such as the locations of agents, the objects currently in the possession of agents, the tools that are accessible within the environment, the ingredients present within each tool, and the tools that are actively in use. Moreover, it includes optional feedback from the environment, triggered when agent actions violate the rules of the environment; for instance, when assigning two distinct actions to the same agent.

**Memory** archives the history of interaction with the environment. Specifically, it chronicles the state of the environment and the state of the agents at every time step.

In addition to the prompt modules, other modules are implemented to help interface between LLMs and CuisineWorld. **Action Extraction** employs a regular expression matching procedure to distill agent actions from the textual output of the LLMs. This module is indispensable because LLM output is not always clean, but may include information reflecting its internal thought processes or even issue apologies for prior missteps in reaction to environmental feedback. **Action Validation** utilizes a look-ahead checking mechanism. This module parses the proposed actions, assessing their feasibility. If an action is deemed unexecutable, an error message is returned.

## 4.1 INFRASTRUCTURE MECHANISMS

Assuming a multi-agent system with $N$ agents, the system must complete a sequence of $P$ different tasks. Each task has $M_p$ different sub-tasks. Furthermore, the number and types of tasks are unknown at the beginning of the episode. The environment will sample a task for the agents to finish during a given interval. The agents must complete the designated task along with other tasks in the task queue. Additionally, each task has an expiration time, after which the task will be marked as a failure. The objective of the multi-agent system is to finish as many tasks as possible and fail as few tasks as possible within a given time frame.

To find optimal task planning, scheduling, and allocations. We define $q_{pim}$ and $c_{pim}$ as quality and cost, respectively, in the context of allocating agent $i$ to work on sub-task $m$ of task $p$ in the episode. For example, if the agent successfully executes an action towards the goal, it will receive a positive quality $q$. For every action, there is an associated cost $c$. Then the combined utility for the sub-task is

$$u_{pim} = \begin{cases} q_{pim} - c_{pim}, & \text{if agent } i \text{ can execute sub-task } m \text{ of task } p \text{ in the episode} \\ -\infty & \text{otherwise.} \end{cases} \tag{2}$$

We define the assignment of sub-task $m$ to agent $i$ as

$$v_{pim} = \begin{cases} 1, & \text{agent } i \text{ is assigned to sub-task } m \text{ of task } p \text{ in the episode} \\ 0 & \text{otherwise.} \end{cases} \tag{3}$$

The goal is to maximize the utility of the episode subject to a time constraint. We define the execution time for task $m$ by agent $i$ for task $p$ in the episode as $\tau_{pim}$, and the maximum time allowed to execute the task as $T_{\max}$, we express the task decomposition and assignment problem as

$$\arg\max_{v} \sum_{p=1}^{P} \sum_{i=1}^{N} \sum_{m=1}^{M_p} u_{pim} v_{pim}, \tag{4}$$

$$\text{subject to} \quad \begin{array}{ll} \sum_p \sum_i \sum_m \tau_{pim} v_{pim} & \leq T_{max} \\ \sum_i v_{pim} & \leq 1 \quad \quad \forall m \in M, \forall p \in P \\ v_{pim} & \in \{0,1\} \quad \forall i \in N, \forall m \in M, \forall p \in P. \end{array} \tag{5}$$

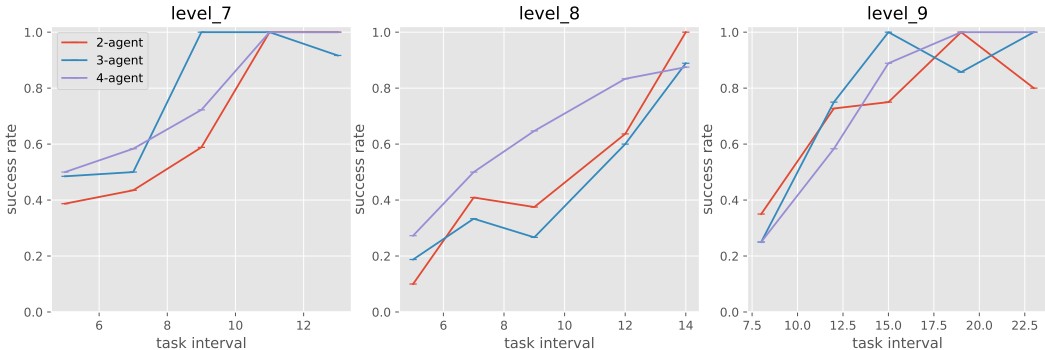

Figure 3: Collaboration efficiency curves on several levels. More level results can be found in the Figure 21 of Appendix E.

| | very simple | | | simple | | | intermediate | | | advanced | | | Average |
|---|---|---|---|---|---|---|---|---|---|---|---|---|---|
| | level 0 | level 1 | level 7 | level 2 | level 4 | level 8 | level 3 | level 9 | level 10 | level 5 | level 11 | level 12 | |
| 2 Agents | 0.727 | 0.706 | 0.682 | **0.687** | **0.664** | 0.504 | 0.764 | 0.725 | 0.701 | 0.661 | 0.692 | 0.559 | 0.673 |
| 3 Agents | **0.781** | **0.778** | **0.780** | 0.528 | 0.600 | 0.455 | 0.822 | **0.771** | **0.815** | 0.689 | **0.733** | **0.570** | **0.694** |
| 4 Agents | 0.771 | 0.761 | 0.761 | 0.505 | 0.592 | **0.626** | **0.848** | 0.744 | 0.790 | **0.692** | 0.675 | 0.534 | 0.692 |

Table 1: Agent **CoS** performance scores on very simple, simple, intermediate, and advanced tasks for various numbers of agents.

Since this problem cannot be solved in polynomial time, we tackle it by leveraging LLMs.

Our prompt design choices try to help an LLM system solve Equation 4. In practice, we reformulate the equation with qualities or rewards expressed in natural language as environmental feedback. For example, when the agent successfully collects an item, the environment emits a signal "collect finish". When the dispatcher assigns a different task to the same agent, the environment emits a signal "agent IDs cannot be the same". As rewards are not immediately observable, we borrow spirits from temporal difference learning. State-action history is accumulated into the memory history. Due to context length limits, it is infeasible to fit the entire history into the context window. We select a fixed horizon history as part of the prompt. We further express the constraints of the system in natural language and repeat important constraints multiple times if necessary.

## 5   EXPERIMENTS AND RESULTS

We have conducted extensive experiments in CuisineWorld. We first introduce the experiment settings and then present an analysis of our empirical results. We report LLM settings in Appendix C. Our experiments focused on addressing the following research questions:
**Q1:** How efficiently can the model dispatch multiple agents?
**Q2:** Can the model dispatch agents for dynamic, on-the-fly goals across different tasks?
**Q3:** How do various components of the input prompt influence the model's performance?
**Q4:** How do other LLMs perform compared to GPT-4?
**Q5:** To what extent can the existing methods collaborate with human users?
**Q6:** What is the human perception of collaborating with numerous intelligent agents?

### 5.1   EXPERIMENTAL REGIMEN I: LLMs DISPATCH MULTI-AGENTS (NPC)

**Collaboration Efficiency (Q1, Q2).** Figure 3 and Table 1 report the performance of our system under different settings. Please find the full results and visualizing figures in Appendix E .

**Findings.** As shown in Figure 3, in general, increasing the number of agents from 2 to 3 will increase the overall performance. However, when increasing more, the overall performance might drop due to the complexity of multi-agent collaborations, as is corroborated by computing **CoS** by levels: As shown in the tables, the **CoS** is the highest when there are two agents in two cases. The **CoS** is the highest when there are three agents in seven cases. The **CoS** is the highest when there are four agents in three cases. Second, we observe that the system performance degrades with more agents under less demanding conditions, indicating that the LLM dispatcher struggles with fewer tasks.

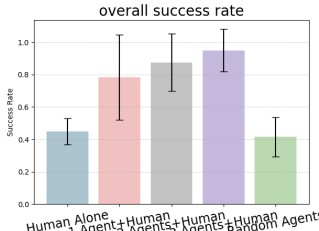 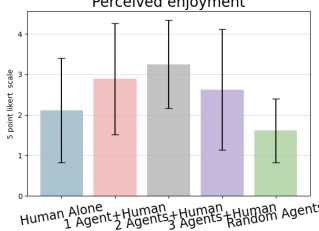 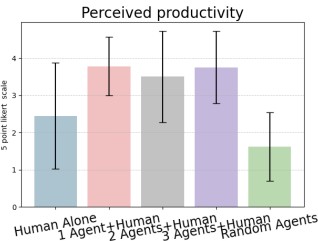

(a) **Collaboration score:** The collaboration score is higher if more agents are collaborating with human players, although the difference is insignificant.

(b) **Perceived enjoyment:** Humans enjoy the game more if they collaborate with the right number of agents.

(c) **Perceived productivity:** Players think collaborating with AI agents will improve productivity.

Figure 4: Human Evaluations. Full results can be found in Figure 27 of Appendix G.3.

## 5.2 EXPERIMENTAL REGIMEN II: HUMAN AND MULTI-NPCS WITH LLMS

### 5.2.1 EXPERIMENTAL SETTING

We conducted a user study in our gaming environment that addresses **Q5** and **Q6**. The user study evaluates the LLM dispatcher's ability to collaborate with humans, where participants are collaborating with 1, 2, and 3 agents or working alone on the virtual cooking tasks. We consider the most general setting, where the LLM works on the unseen task, as Level_3.

### 5.2.2 EXPERIMENT DESIGN

**Hypotheses.** The user study tests the following hypotheses:

- **H1: Task productivity**. Participants have higher productivity when collaborating with AI agents.
- **H2: Task productivity with more agents**. Participants have higher productivity when collaborating with more AI agents.
- **H3: Perception of the AI agents.** Participants have higher perceived task efficiency and more fun playing the game as a consequence of the collaboration.

**Manipulated Variables.** We use a within-subject design for our experiment. Every user tries to finish the task solo or collaborates with different numbers of agents with varying competency. We randomize the order of the treatment to mitigate practice effects, fatigue effects, and carryover effects.

- **Single agent:** Participants work on the task by themselves.
- **LLM-powered multi-agent system:** Participants collaborate with the multi-agent AI system powered by an LLM.
- **Random agent:** Random agents execute random actions from a pool of valid actions. Participants collaborate with random AI agents.

We recruited 12 subjects for our study, including 2 females and 10 males. We used ANOVA to test the effects of different experimental conditions on collaboration performance and the subjective perceptions. Tukey HSD tests were conducted on all possible pairs of experimental conditions.

**Findings.** As showed in Figure 4 and Figure 27, we found significant effects on the team collaboration success rate $F(4, 55) = 28.11$, $p < 0.001$. Post-hoc comparisons using Tukey HSD tests revealed that the team comprising the human player with LLM agents achieves a higher success rate than the human working alone ($p < 0.001$) across different numbers of agents, **thus confirming H1**. Although collaborating with more agents had a higher success rate, it was not significantly different from collaborating with one, two, or three agents ($p = 0.774$ and $p = 0.231$, respectively). We observed that human players have more fun playing the game when collaborating with LLM-powered AI agents than when playing alone ($p = 0.0126$). Players felt that collaboration with AI agents leads to higher productivity ($p = 0.0104$), **thus confirming H3**. Additionally, when playing with AI agents, human players take their actions based on other players' actions ($p = 0.00266$). Human players also found that AI agents are more predictable than random agents ($p < 0.001$). Further insights from player feedback highlighted an intriguing trade-off: while greater numbers of agents improved overall task success rates, this reduced the enjoyment of the game. Often, players felt sidelined and

| 2 agent | GPT-4 (full) | GPT-4 w/ only few-step | GPT-4 w/o inference knowledge | GPT-4 w/o feedback |
|---------|--------------|------------------------|-------------------------------|--------------------|
| **CoS** | 0.764 | 0.710 | 0.714 | 0.311 |

Table 2: Additional ablation on Level 3

| level_3 | 4agent using 4agent demo | 4agent using 2agent demo | 3agent using 3agent demo | 3agent using 2agent demo |
|---------|--------------------------|--------------------------|--------------------------|--------------------------|
| **CoS** | 0.848 | 0.851 | 0.822 | 0.775 |

Table 3: Using different numbers of agents as one-shot demonstrations

less involved. Thus, game developers should adjust AI performance to maintain player engagement and fun. As suggested by Yuan et al. (2022), aligning human values with AIs is a promising approach. The visualization figures of CuisineWorld showed in Appendix E.2.

# 6 ANALYSIS AND EMERGENT GAMING ABILITIES

## 6.1 ABLATION STUDY FOR MULTI-AGENTS

**Study of the Prompt Components (Q3).** In Table 2, we elucidate the performance of LLM dispatchers with certain components of the prompt omitted. We chose level 3 as the basis for our ablation study. It was selected because it displayed a clear correlation between an increased number of agents and improved performance, serving as a stable benchmark for evaluating the LLM's coordination capabilities. We recognize that the LLM may not perform consistently across all levels on the challenging CuisineWorld benchmark. When performance is variable or poor on certain levels, it can obscure the effects of ablation studies due to multiple confounding factors. Details about the prompt can be found in the appendices. Specifically, for these tests, we excluded individual components such as the inference knowledge, reduced the prompt example to a mere two steps instead of the complete demonstration, and evaluated the model without environmental feedback.

**Findings.** Table 2 indicates a significant drop in performance when environmental feedback is excluded, underscoring its pivotal role in the efficacy of the LLM dispatcher. Replaying action sequences reveals that, without feedback, the LLM dispatcher tends to repeat mistakes and gets stuck in specific states for prolonged durations. Another key takeaway is that a succinct two-step demonstration of input and output format can still achieve impressive performance for unseen tasks with dynamic objectives. Notably, in these two-step instances, there is no explicit guide to finishing any tasks, yet the model does not merely complete the task but continually performs additional tasks within the same episode. Furthermore, we observe that integrating human-crafted inference knowledge bolsters the performance of the LLM dispatcher. Lastly, even with few-shot demonstrations involving fewer agents, the LLM dispatcher retains satisfactory performance, as shown in Table 3. Despite some impressive findings, we also observe several drawbacks through our experiments. 1) GPT-4 heavily relies on feedback, without feedback its performance drops significantly as indicated in our Table 2. 2) GPT-4 is sensitive to prompt inputs as indicated in Table 2, without inference knowledge, the performance will drop.

**Study of the Performance of other LLMs (Q4).** To study how other LLMs perform on our tasks, we tested the collaboration performance of GPT-3.5, Claude-2, and LLaMA2, and Table 4 summarizes the results. For a fair comparison, all tests employed identical prompt inputs.

**Findings.** We observed that while other LLMs tend to underperform, models such as Claude-2 still manage to complete the task to a considerable extent.

## 6.2 EMERGENT ABILITIES

Across our experiments, our MindAgent framework exhibits the following emergent properties:

**Emergent Collaboration Task Understanding.** As shown in Table 2, especially in the few-step ablation entries, GPT-4 exhibits its proficiency even when not provided with a full demonstration

| | 2 agents | | | | 3 agents | | | | 4 agents | | | |
|---|-------|----------|--------|---------|-------|----------|--------|---------|-------|----------|--------|---------|
| | GPT-4 | Claude-2 | LLaMA2 | ChatGPT | GPT-4 | Claude-2 | LLaMA2 | ChatGPT | GPT-4 | Claude-2 | LLaMA2 | ChatGPT |
| **CoS** | 0.686 | 0.3125 | 0 | 0 | 0.822 | 0.372 | 0 | 0 | 0.848 | 0.473 | 0 | 0 |

Table 4: Performance of other LLMs on Level 3

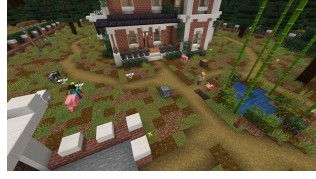 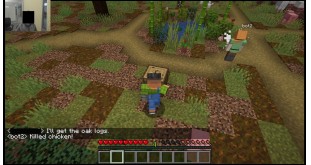 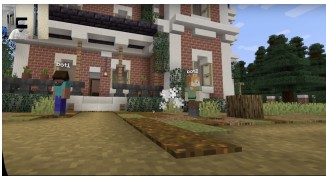

|        (a) Multi-Agent        |        (b) Human-NPC        |        (c) VR        |

Figure 5: (a) Alex and Steve are collaborating to kill different animals. (b) A human player instructs the agents to perform certain actions. (c) A human player collaborating with agents in VR.

| GPT-4 Minecraft | $\tau_{\text{int},(1)}$ | $\tau_{\text{int},(2)}$ | $\tau_{\text{int},(3)}$ | $\tau_{\text{int},(4)}$ | $\tau_{\text{int},(5)}$ | **CoS** |
|---|---|---|---|---|---|---|
| Performance | 0.195 | 0.381 | 0.704 | 0.792 | 0.833 | 0.581 |

Table 5: Performance of MindAgent framework in Minecraft

of specific tasks. To clarify, a "full few-shot demo" typically refers to a comprehensive demonstration of a task, detailing each step and procedure involved. By contrast, we provide GPT-4 with only a partial demonstration or a glimpse of the task executing only two steps. Yet, despite this limited input, GPT-4's performance is remarkable. This underscores GPT-4's impressive **emergent zero-shot multi-agent planning** abilities. Beyond simply completing unseen tasks, GPT-4 also demonstrates adaptability by dynamically prioritizing multiple different tasks as they arise, emphasizing its **emergent multi-task, on-the-fly planning** skills.

**Emergent Multi-agent Reasoning Abilities.** Referring to Table 3, GPT-4 has the ability to deploy more agents based on demonstrations of fewer agents. For instance, it can effectively dispatch 4 agents having only seen demonstrations involving 2 agents. However, performance is better when 4 agents use 2-agent demos compared to 4-agent demos, possibly because the task suits 2 or 4-agent teams. With 4 agents, the model can create two independent teams of two, showcasing its ability to allocate tasks and plan for more agents than previously experienced.

## 7 NOVEL GAME ADAPTATION

In line with our ongoing efforts to create collaborative, in-game, multi-agent systems, we integrated our infrastructure into Minecraft (Figure 5). In this adaptation, we designed several unique cooking tasks where two in-game agents, Alex and Steve, must cook various types of meat as shown in Appendix F.2. After cooking, they must deposit the meats into a chest. See Table 5 for the experimental results, and see Appendix F.3 for additional visualizations. The action details of Minecraft please take the reference in Appendix F.4. We provide more details for transfering to Minecraft in F.1.

Incorporating game-specific domain knowledge into the system is crucial to games where domain-specific knowledge plays an important part. In CuisineWorld and Minecraft, we inject domain-specific knowledge (such as recipes) directly into the context, which the LLMs utilize to inform the decision-making and collaboration strategies. This demonstrates the feasibility of adapting MindAgent to other domains where specific knowledge plays a crucial role. In addition, techniques like Retrieval-Augmented Generation (RAG) and Fine Tuning could be pivotal in further developing MindAgent's capabilities to handle such domain-specific complexities.

## 8 CONCLUSION

We have introduced MindAgent, an infrastructure for multi-agent collaboration through LLMs across multiple gaming domains. We investigated its multi-agent planning capabilities, and we deployed our infrastructure into real-world video games that demonstrate its multi-agent and human-AI collaboration effectiveness. Additionally, we presented CuisineWorld, a text-based multi-agent collaboration benchmark that provides a new auto-metric Collaboration Score (CoS) to quantify collaboration efficiency. In this work, we mainly introduce the interpretable human-NPCs communication in collaborative games, which is a promising direction for enhancing multi-agents cooperation as the interactive way forward. Beyond its practical applications, we anticipate that our work will guide the development of future gaming systems in which human-AI collaboration is seamless and intuitive. We are optimistic that our insights and findings will catalyze the design of games that are both technologically advanced and significantly more engaging and enjoyable for players.

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

# Appendix for
# MindAgent: Emergent Gaming Interaction

## A  PROMPT EXAMPLES

We provide some examples of prompts for CuisineWorld. Figure 6 shows an example of the system prompt info. Figure 7 shows an example of a partial demonstration.

```
The available actions are :
1) goto: goto a tool location
2) get: get some object from a tool
3) put: put some object into a tool
4) activate: activate the tool to cook all ingredients inside the tool into a different tools
5) noop: not performing any actions
Sometimes the system will give you error messages. Please consider these error messages when executing actions.
You need to specify action for all of the agents, **except human**. They all have different agent numbers. Do not assign actions to the same agent more than once.

When the tools reach its capacity, you need to take stuff out. Otherwise, you cannot put items inside.
When you are holding objects, you cannot get any more objects.
When you are holding objects, you cannot activate tools.
After you cooked a required dish, you need to put it into the servingtable.
You can only pick up objects from the tool location, if you are located at the tool location.
When you activate any tools, make sure all the items inside the tool are respecting the recipes. Otherwise, you will cook waste. Avoid waste at all cost.
*** You should mix salad in the mixer. To make salad you should chop veggies first. ***
*** If the tool is occupied, indicated by the occupy() predicate, you cannot get objects from it or put objects into it. ***
*** The food orders are keep coming. You should finish as many dishes as possible and finish every dish as soon as possible. Please deliver the order to the serveringtable when it is finished. ***
*** The dish will expire after the lifetime reaches 0 and it's not at the serveringtable. Please avoid this. *** Here are the recipes:

Cook porkMeatcake at:
 -- location: blender
 -- with ingredients:      pork,     flour,
Cook salmonSashimi at:
 -- location: chopboard
 -- with ingredients:      salmon,
Cook tunaSashimi at:
 -- location: chopboard
 -- with ingredients:      tuna,
Cook mixedSashimi at:
 -- location: mixer
 -- with ingredients:      salmonSashimi,    tunaSashimi,
The following objects are available:
 --1) salmonSashimi
 --2) tuna
 --3) mixedSashimi
 --4) tunaSashimi
 --5) porkMeatcake
 --6) salmon
 --7) flour
 --8) pork
The objects are cooked using tools or are just base ingredients.
Among them, the following are base ingredients:
 --1) tuna
 --2) salmon
 --3) flour
 --4) pork
You can only obtain base ingredients from the storage initially.
Additional rules:
You can place up to infinite item into the storage0
You can place up to infinite item into the storage0
You can place up to infinite item into the servingtable0
You can place up to infinite item into the servingtable0
You can place up to 1 item into the chopboard0
You can place up to 1 item into the chopboard0
You can place up to 1 item into the chopboard1
You can place up to 1 item into the chopboard1
You can place up to 5 item into the mixer0
You can place up to 5 item into the mixer0
You can place up to 5 item into the mixer1
You can place up to 5 item into the mixer1
** Only ** the following tools are available:
storage0, servingtable0, chopboard0, chopboard1, mixer0, mixer1, You cannot pick up these tools. You can only use those tools at the corresponding location.
```

Figure 6: The MindAgent system prompt example.

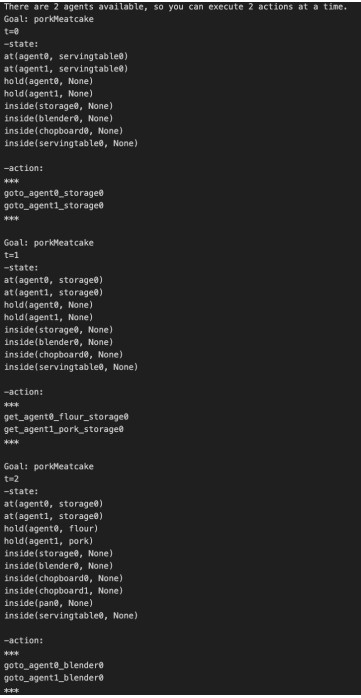

Figure 7: The MindAgent system partial one-shot demo example.

## B  PROMPT ENGINEERING DETAILS

Our initial approach began with a simple and generic prompt, designed to be as neutral as possible. This baseline prompt was tested across all models, including GPT-4, Claude, and other LLMs in our study. We observed that only GPT-4 and Claude were able to generate reasonable planning responses with this initial prompt. Consequently, we refined our prompt engineering efforts to better suit these two models, aiming to optimize their performance and response quality.

For the other LLMs, we also attempted individual prompt engineering. However, these efforts did not yield significant improvements in their responses compared to the initial trial. After several rounds of testing and refinement, we concluded that further prompt customization for these models did not result in notably better outcomes.

As a result, we decided to use a consistent prompt across all LLMs for the final comparison. This decision was made to maintain uniformity in testing conditions, despite the prompts being more closely tuned to GPT-4 and Claude. We believe this approach offers a reasonable balance between fairness and practicality in evaluating the capabilities of different LLMs under similar conditions.

## C  LLM SETTINGS

We perform experiments on CuisineWorld through OpenAI APIs and Anthropic APIs. All GPT-4 experiments employ the gpt-4-0613 model, and all Chat-GPT experiments employ gpt-3.5-turbo-0613. For the Llama 2 experiments, we use the hugging face inference endpoints Llama-2-70b-chat-hf. We set the temperature for all experiments to 0.1 following Wang et al. (2023a). We report the average results over three episodes.

## D  CUISINEWORLD TASK DETAILS

### D.1  CUISINEWORLD TASK DEFINITIONS

We follow prior work (Yao et al., 2023; Liu et al., 2023; Deng et al., 2023) to **interactively evaluate LLMs as planning agents**. Overall, the interactive evaluation can be formulated as a *Markov Decision Process* $(\mathcal{S}, \mathcal{A}, \mathcal{T}, \mathcal{R}, \mathcal{G})$, with state space $\mathcal{S}$, action space $\mathcal{A}$ (effectively indicating all the possible schedules that can be made at a single time step), transition dynamics $\mathcal{T}$, reward function $\mathcal{R}$, and task instruction space $\mathcal{G}$. Note that, although there are multiple agents inside CuisineWorld that can be coordinated, asmentioned above, we adopt a centralized planning scheme and thereby formulate our game as a single-agent, fully-observable decision-making problem. An illustration of the state & action space and the possible tasks of our game can be found in Figure 1 of the paper.

**State Space $\mathcal{S}$.** In a CuisineWorld virtual kitchen, there are two types of entities: `location` and `agent`. For each entity, the game will provide a set of descriptions, and the aggregated descriptions of all entities will be the state returned by the game. A `location` can be *storage*, where one can obtain ingredients and dispense waste, a *serving table*, onto which one should put the completed, or a cooking tool; *e.g.*, *pan* or *blender*. We offer up to two descriptions for each location: `inside(location, items)`, indicating what items (some ingredients, completed dishes, *etc*.) are now inside the location, and `occupy(location)`, suggesting `location` is now being used and cannot be touched; *e.g.*, an activated blender. An `agent` is an entity that can be dispatched to complete the task, and we provide up to three descriptions for each agent: `at(location, agent)`, indicating that `agent` is now at `location`, `hold(agent, items)`, suggesting what items `agent` is holding, and `occupy(agent)`, implying `agent` is now operating a tool, *e.g.*, chopping some fruits, and will not respond to any dispatching command. The set of tool distributions can be found in Table 23.

**Action Space $\mathcal{A}$.** An action in CuisineWorld is a list of dispatching commands. Given $N$ `agent` entities, a total of $N$ commands must be generated. The agent provides the following commands (also tabulated in Table 14):

1. `goto(agent, location)`, to let `agent` move to `location`;

2. `get(agent, location, item)`, to let `agent` get a specific item from `location`;

3. `put(agent, location)`, to put whatever `agent` is holding into `location`;

4. `activate(agent, location)`, to let `agent` turn on `location` if it is a cooking tool, *e.g.* *blender*;

5. `noop(agent)`, to have `agent` perform no actions in this round of dispatching.

Note that, to avoid the possible confusion of multiple agents being dispatched to operate with the same `location`, the dispatcher also must properly order the dispatching commands as they will be executed sequentially.

## D.2   IMPLEMENTING CUISINEWORLD

The implementation of CuisineWorld mostly follows the spirit of *Overcooked!*, a renowned video game. Therefore, we refer to many of its game mechanisms while simplifying some of them; *e.g.*, we skip low-level control and assume all `agent` entities have access to all `location` at any time. Specifically, we crawled the rules and recipes from the community-contributed wiki[1] of *Overcooked!* streamlined them, and made necessary modifications, ending up with the basic version of CuisineWorld comprising **10** types of `location` (*serving table*, *storage*, and 8 different cooking tools), **27** types of ingredients, and **33** unique dishes. We grouped the dishes based on their difficulty (primarily based on the number of cooking tools involved) to design and implement **12** game levels, which are further categorized into 4 classes: *entry*, *simple*, *intermediate*, and *advanced*, with 3 levels each. Note that the recipes, dishes, and levels can be easily extended to incorporate more challenging tasks.

## D.3   TASK GRAPH VISUALIZATION

In CuisineWorld, we provide tasks of different complexities to holistically evaluate the multi-agent system's performance. Additionally, the environment is highly customizable and extendable. Users only need only modify the JSON files to add more tasks or modify existing tasks. In the following sebsections, we visualize different CuisineWorld task graphs.

### D.3.1   LEVEL 0 – VERY SIMPLE

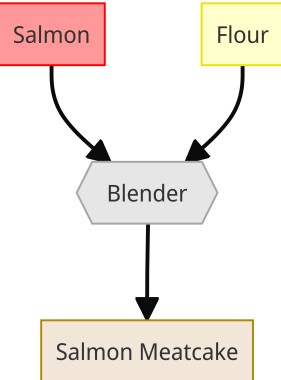

Figure 8: Salmon Meatcake

---

[1]https://steamcommunity.com/sharedfiles/filedetails/?id=1769729191

### D.3.2 LEVEL 1 – VERY SIMPLE

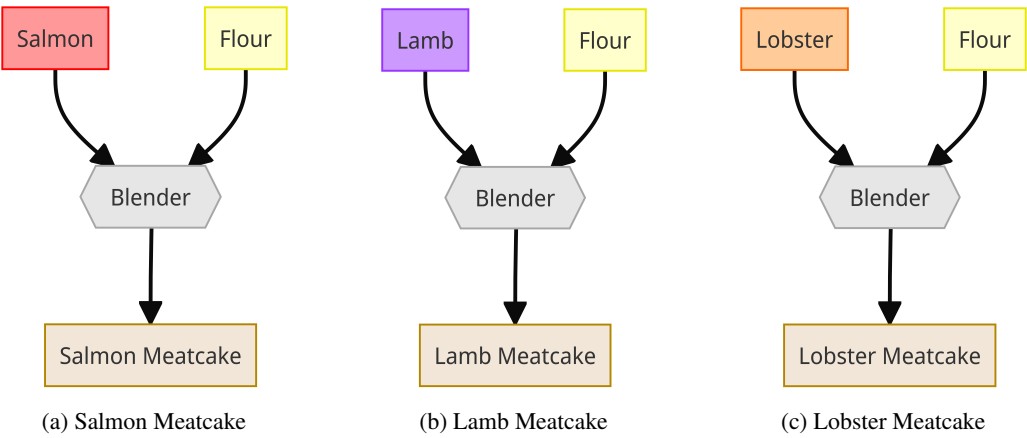

(a) Salmon Meatcake  (b) Lamb Meatcake  (c) Lobster Meatcake

### D.3.3 LEVEL 2 – SIMPLE

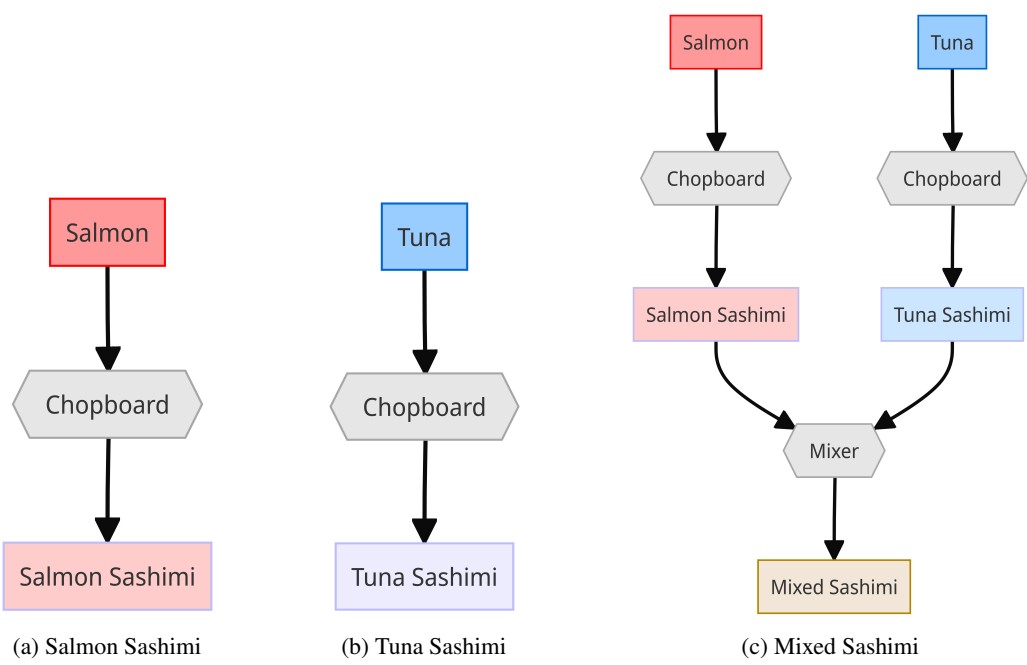

(a) Salmon Sashimi  (b) Tuna Sashimi  (c) Mixed Sashimi

### D.3.4    LEVEL 3 – INTERMEDIATE

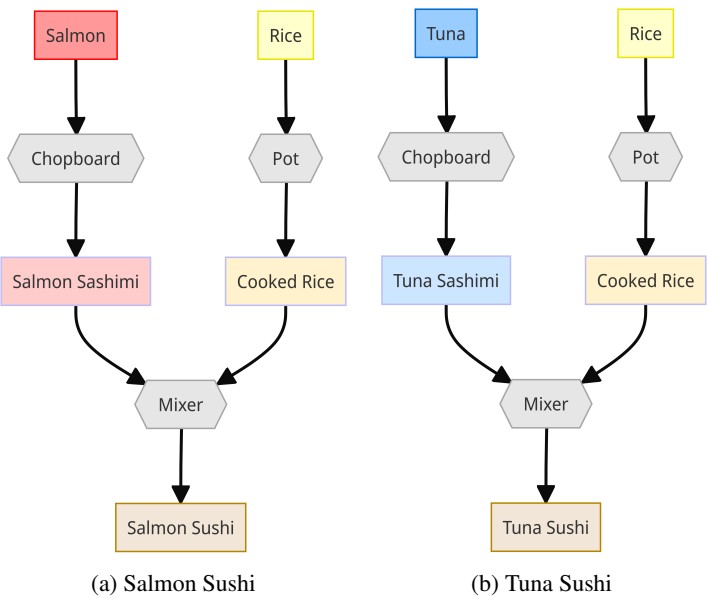

(a) Salmon Sushi          (b) Tuna Sushi

### D.3.5    LEVEL 4 – SIMPLE

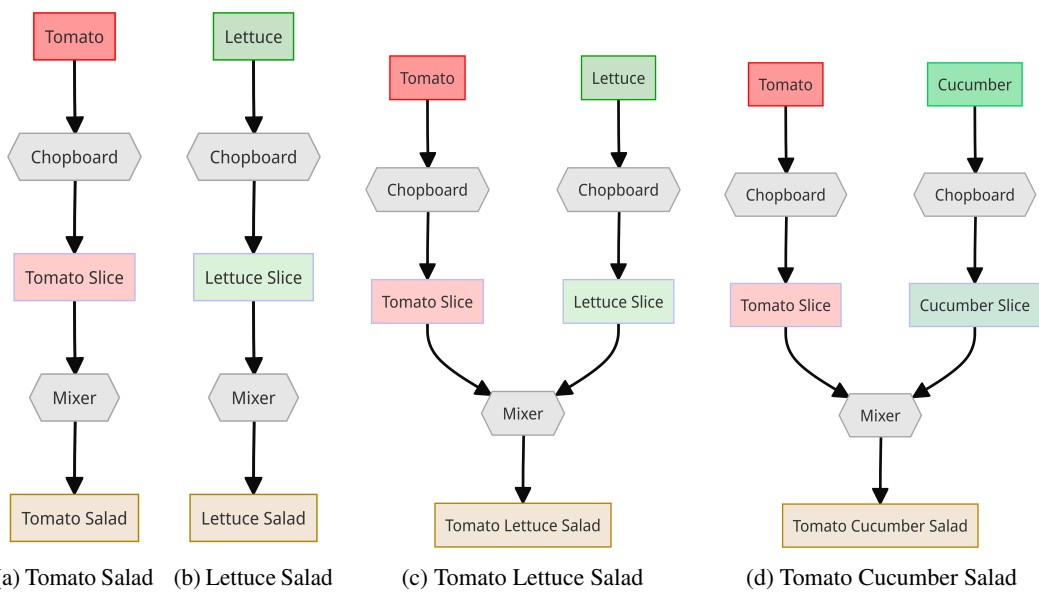

(a) Tomato Salad    (b) Lettuce Salad    (c) Tomato Lettuce Salad    (d) Tomato Cucumber Salad

### D.3.6  LEVEL 5 – ADVANCED

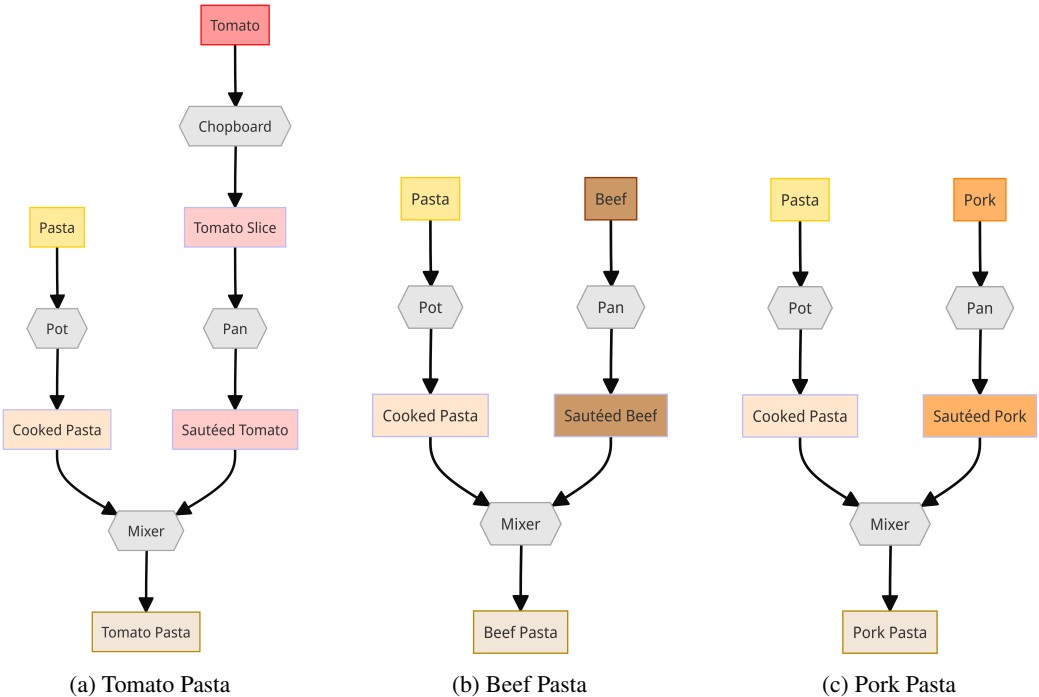

(a) Tomato Pasta                (b) Beef Pasta                (c) Pork Pasta

### D.3.7  LEVEL 6 – UNUSED

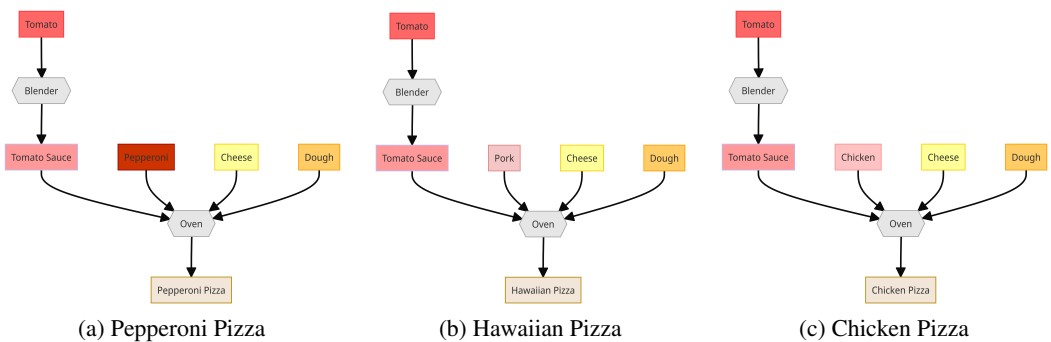

(a) Pepperoni Pizza             (b) Hawaiian Pizza           (c) Chicken Pizza

### D.3.8  LEVEL 7 – VERY SIMPLE

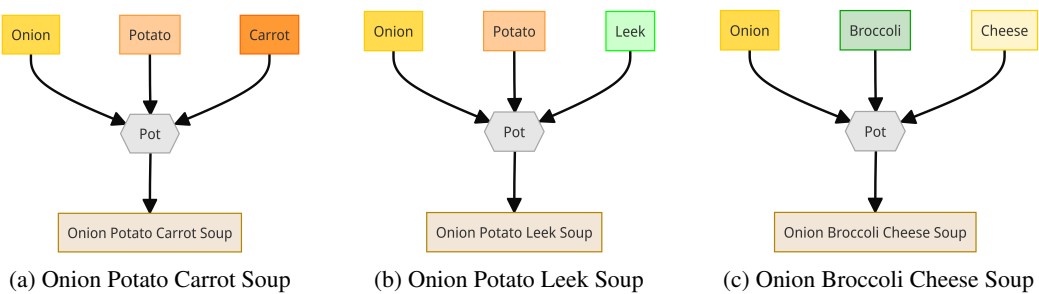

(a) Onion Potato Carrot Soup    (b) Onion Potato Leek Soup   (c) Onion Broccoli Cheese Soup

### D.3.9 LEVEL 8 – SIMPLE

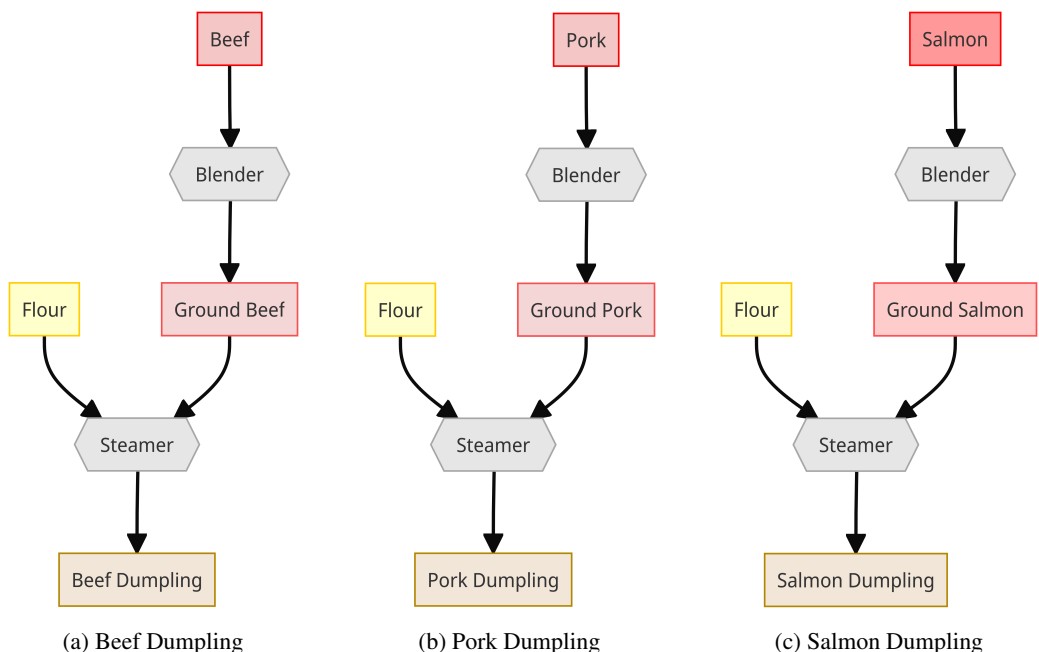

(a) Beef Dumpling     (b) Pork Dumpling     (c) Salmon Dumpling

### D.3.10 LEVEL 9 – INTERMEDIATE

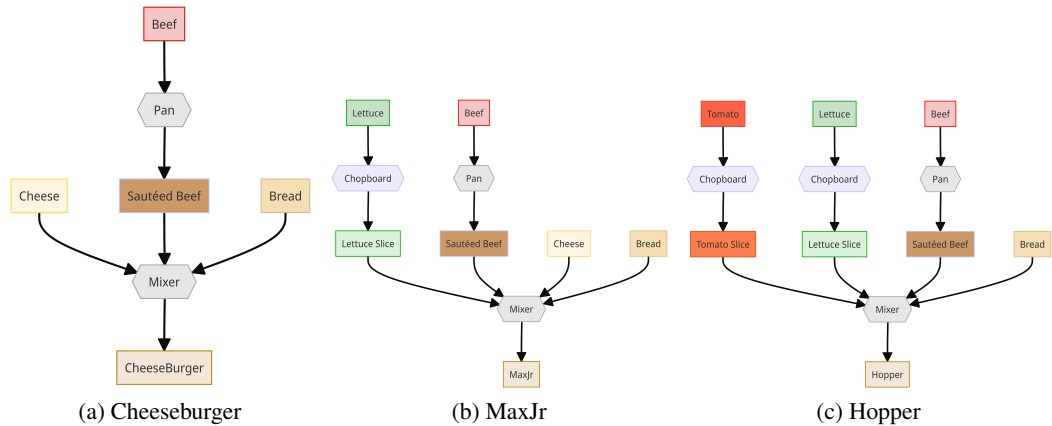

(a) Cheeseburger     (b) MaxJr     (c) Hopper

### D.3.11 LEVEL 10 – INTERMEDIATE

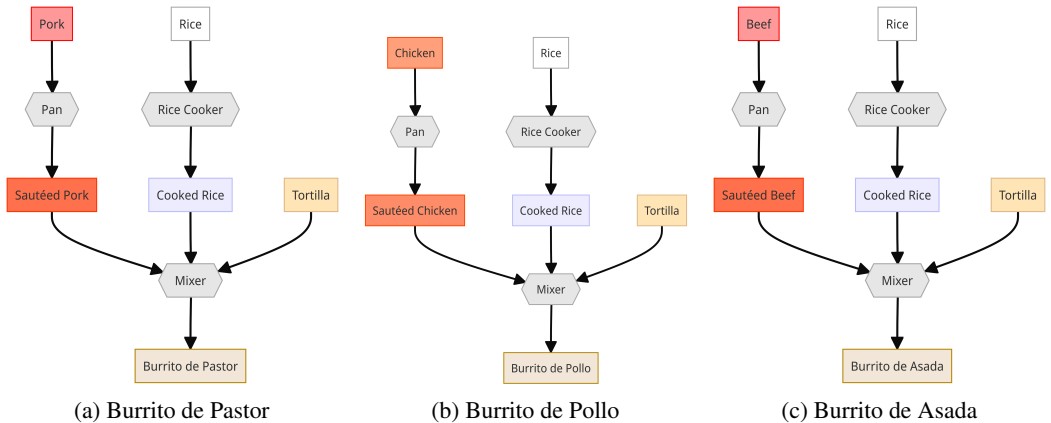

(a) Burrito de Pastor      (b) Burrito de Pollo      (c) Burrito de Asada

### D.3.12 LEVEL 11 – ADVANCED

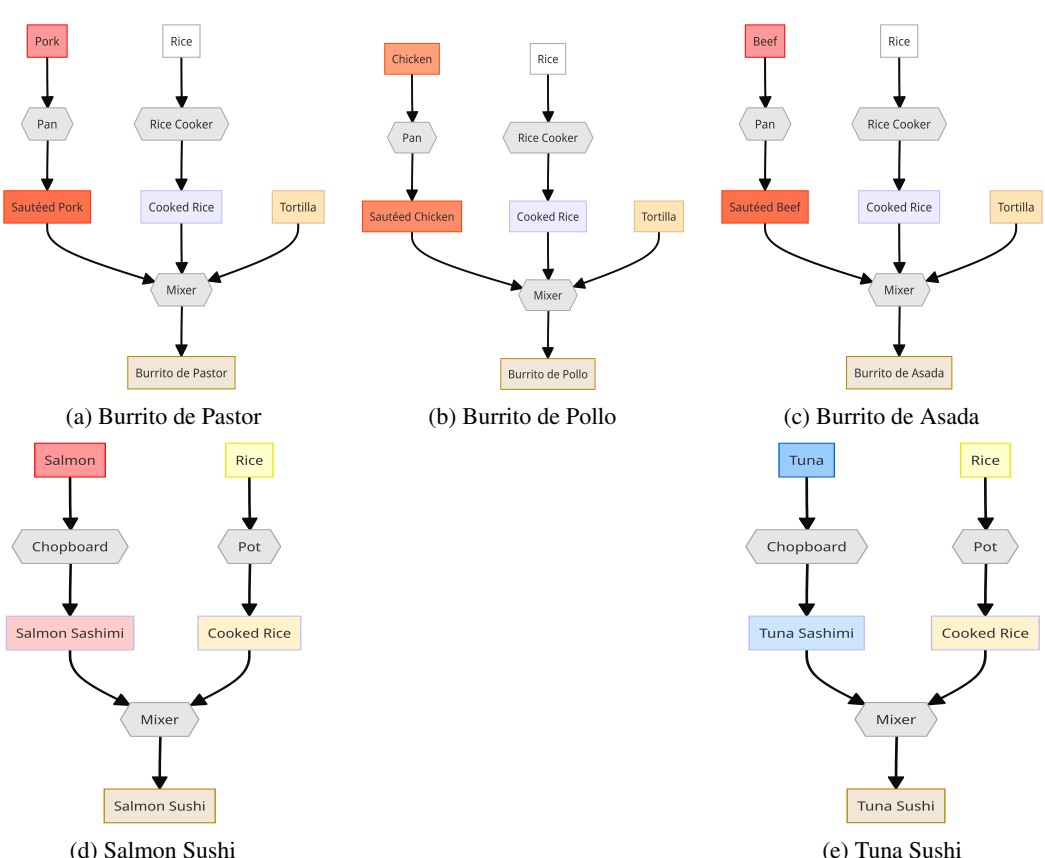

(a) Burrito de Pastor      (b) Burrito de Pollo      (c) Burrito de Asada

(d) Salmon Sushi      (e) Tuna Sushi

### D.3.13 LEVEL 12 – ADVANCED

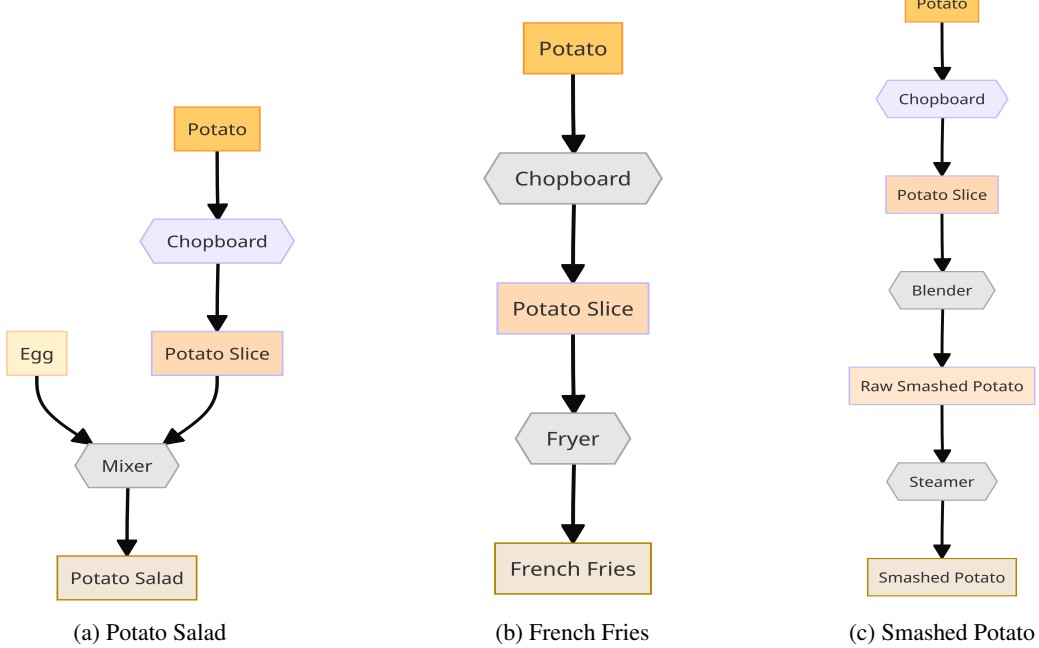

(a) Potato Salad      (b) French Fries      (c) Smashed Potato

# E   ADDITIONAL RESULTS IN CUISINEWORLD

The following tables report additional performance results for several different numbers of agents and task complexity levels, performance of other LLMs, and additional ablation results.

| 2-agent | very simple | | | simple | | | intermediate | | | advanced | | | Avg. |
|---|---|---|---|---|---|---|---|---|---|---|---|---|---|
| | level 0 | level 1 | level 7 | level 2 | level 4 | level 8 | level 3 | level 9 | level 10 | level 5 | level 11 | level 12 | |
| GPT4 $\tau_{\text{int},(1)}$ | 18/54 | 18/56 | 12/31 | 14/34 | 12/30 | 3/30 | 10/26 | 7/20 | 7/23 | 6/23 | 6/21 | 10/36 | 0.318 |
| GPT4 $\tau_{\text{int},(2)}$ | 18/31 | 17/34 | 10/23 | 13/26 | 12/22 | 9/22 | 10/17 | 8/11 | 6/12 | 5/13 | 4/14 | 8/21 | 0.486 |
| GPT4 $\tau_{\text{int},(3)}$ | 18/25 | 19/25 | 10/17 | 16/18 | 11/18 | 6/16 | 11/13 | 6/8 | 7/10 | 8/10 | 9/9 | 8/17 | 0.709 |
| GPT4 $\tau_{\text{int},(4)}$ | 18/18 | 18/19 | 12/12 | 11/14 | 11/12 | 7/11 | 12/12 | 8/8 | 9/9 | 6/7 | 8/9 | 11/12 | **0.912** |
| GPT4 $\tau_{\text{int},(5)}$ | 18/18 | 17/17 | 12/12 | 11/13 | 11/13 | 9/9 | 11/11 | 4/5 | 7/7 | 8/8 | 8/8 | 9/12 | **0.937** |
| **CoS** | 0.727 | 0.706 | 0.682 | **0.687** | **0.664** | 0.504 | 0.764 | 0.725 | 0.701 | 0.661 | 0.692 | 0.559 | 0.673 |

Table 6: 2 agents performance on different tasks

| 3-agent | very simple | | | simple | | | intermediate | | | advanced | | | Average |
|---|---|---|---|---|---|---|---|---|---|---|---|---|---|
| | level 0 | level 1 | level 7 | level 2 | level 4 | level 8 | level 3 | level 9 | level 10 | level 5 | level 11 | level 12 | |
| GPT4 $\tau_{\text{int},(1)}$ | 21/55 | 24/55 | 16/33 | 17/33 | 9/28 | 6/32 | 12/25 | 5/20 | 8/21 | 7/22 | 7/22 | 9/26 | **0.368** |
| GPT4 $\tau_{\text{int},(2)}$ | 20/31 | 25/33 | 11/22 | 4/24 | 13/24 | 7/21 | 14/20 | 9/12 | 9/13 | 7/14 | 8/14 | 10/23 | 0.549 |
| GPT4 $\tau_{\text{int},(3)}$ | 22/25 | 21/26 | 17/17 | 11/20 | 9/17 | 4/15 | 13/14 | 8/8 | 12/12 | 7/7 | 9/10 | 10/16 | **0.791** |
| GPT4 $\tau_{\text{int},(4)}$ | 22/22 | 20/21 | 14/14 | 9/13 | 7/10 | 6/10 | 10/10 | 6/7 | 10/10 | 5/8 | 7/8 | 11/13 | 0.846 |
| GPT4 $\tau_{\text{int},(5)}$ | 20/20 | 15/16 | 11/12 | 10/14 | 10/11 | 8/9 | 12/12 | 6/6 | 8/8 | 5/5 | 8/8 | 6/10 | 0.914 |
| CoS | **0.781** | **0.778** | **0.780** | 0.528 | 0.600 | 0.455 | 0.822 | **0.771** | **0.815** | 0.689 | **0.733** | **0.570** | **0.694** |

Table 7: 3 agents performance on different tasks

| 4-agent | very simple | | | simple | | | intermediate | | | advanced | | | Average |
|---|---|---|---|---|---|---|---|---|---|---|---|---|---|
| | level 0 | level 1 | level 7 | level 2 | level 4 | level 8 | level 3 | level 9 | level 10 | level 5 | level 11 | level 12 | |
| GPT4 $\tau_{\text{int},(1)}$ | 22/54 | 18/55 | 17/34 | 13/34 | 8/28 | 9/33 | 16/27 | 5/20 | 8/23 | 5/22 | 8/22 | 8/35 | 0.349 |
| GPT4 $\tau_{\text{int},(2)}$ | 24/32 | 21/33 | 14/24 | 14/25 | 12/24 | 11/22 | 16/19 | 7/12 | 9/15 | 7/14 | 6/12 | 12/23 | **0.590** |
| GPT4 $\tau_{\text{int},(3)}$ | 23/25 | 23/26 | 13/18 | 11/19 | 10/17 | 11/17 | 15/17 | 8/9 | 11/11 | 7/8 | 10/11 | 9/17 | 0.785 |
| GPT4 $\tau_{\text{int},(4)}$ | 22/22 | 21/22 | 14/14 | 7/15 | 10/13 | 10/12 | 12/13 | 9/9 | 10/10 | 6/7 | 8/8 | 9/13 | 0.875 |
| GPT4 $\tau_{\text{int},(5)}$ | 14/18 | 20/20 | 14/14 | 7/13 | 9/11 | 7/8 | 12/12 | 5/5 | 7/7 | 6/6 | 3/5 | 7/10 | 0.859 |
| CoS | 0.771 | 0.761 | 0.761 | 0.505 | 0.592 | **0.626** | **0.848** | 0.744 | 0.790 | **0.692** | 0.675 | 0.534 | 0.692 |

Table 8: 4 agents performance on different tasks

| | 2 agent | | | | 3 agent | | | | 4 agent | | | |
|---|---|---|---|---|---|---|---|---|---|---|---|---|
| | GPT-4 | Claude-2 | LLaMA | ChatGPT | GPT-4 | Claude-2 | LLaMA | ChatGPT | GPT-4 | Claude-2 | LLaMA | ChatGPT |
| $\tau_{\text{int},(1)}$ | 10/26 | 3/24 | 0 | 0/24 | 12/25 | 5/26 | 0 | 0/24 | 16/27 | 9/25 | 0 | 0/24 |
| $\tau_{\text{int},(2)}$ | 10/17 | 3/16 | 0 | 0/15 | 14/20 | 4/16 | 0 | 0/15 | 16/19 | 4/15 | 0 | 0/15 |
| $\tau_{\text{int},(3)}$ | 11/18 | 3/12 | 0 | 0/12 | 13/14 | 3/12 | 0 | 0/12 | 15/17 | 4/12 | 0 | 0/12 |
| $\tau_{\text{int},(4)}$ | 11/13 | 3/9 | 0 | 0/9 | 10/10 | 5/11 | 0 | 0/9 | 12/13 | 6/11 | 0 | 0/9 |
| $\tau_{\text{int},(5)}$ | 11/11 | 4/6 | 0 | 0/6 | 12/12 | 5/7 | 0 | 0/6 | 12/12 | 6/7 | 0 | 0/6 |
| CoS | 0.686 | 0.3125 | 0 | 0 | 0.822 | 0.372 | 0 | 0 | 0.848 | 0.473 | 0 | 0 |

Table 9: Performance of other LLMs on Level 3

| 2 agent | GPT-4 | GPT-4 w/ few-step | GPT-4 w/o inference knowledge | GPT-4 w/o feedback |
|---|---|---|---|---|
| $\tau_{\text{int},(1)}$ | 10/26 | 8/26 | 8/25 | 4/25 |
| $\tau_{\text{int},(2)}$ | 10/17 | 11/19 | 9/17 | 4/17 |
| $\tau_{\text{int},(3)}$ | 11/13 | 11/13 | 10/12 | 4/12 |
| $\tau_{\text{int},(4)}$ | 12/12 | 9/11 | 8/9 | 1/9 |
| $\tau_{\text{int},(5)}$ | 11/11 | 10/10 | 9/9 | 5/7 |
| CoS | 0.764 | 0.710 | 0.714 | 0.311 |

Table 10: Additional ablation results

| level_3 | 4agent using 4agent demo | 4agent using 2agent demo | 3agent using 3agent demo | 3agent using 2agent demo |
|---|---|---|---|---|
| GPT4 $\tau_{\text{int},(1)}$ | 16/27 | 14/27 | 12/25 | 11/25 |
| GPT4 $\tau_{\text{int},(2)}$ | 16/19 | 16/20 | 14/20 | 11/19 |
| GPT4 $\tau_{\text{int},(3)}$ | 15/17 | 15/16 | 13/14 | 12/14 |
| GPT4 $\tau_{\text{int},(4)}$ | 12/13 | 13/13 | 10/10 | 12/12 |
| GPT4 $\tau_{\text{int},(5)}$ | 12/12 | 12/12 | 12/12 | 11/11 |
| CoS | 0.848 | 0.851 | 0.822 | 0.775 |

Table 11: Using different numbers of agents demos

| model | level 1 | level 4 |
|---|---|---|
| RL Performance | 0.451 | 0.598 |
| Ours(GPT4 $\tau_int(4)$) | 0.947 | 0.917 |

Table 12: Performance of masked PPO with 2 agents in level 1 and level 4.

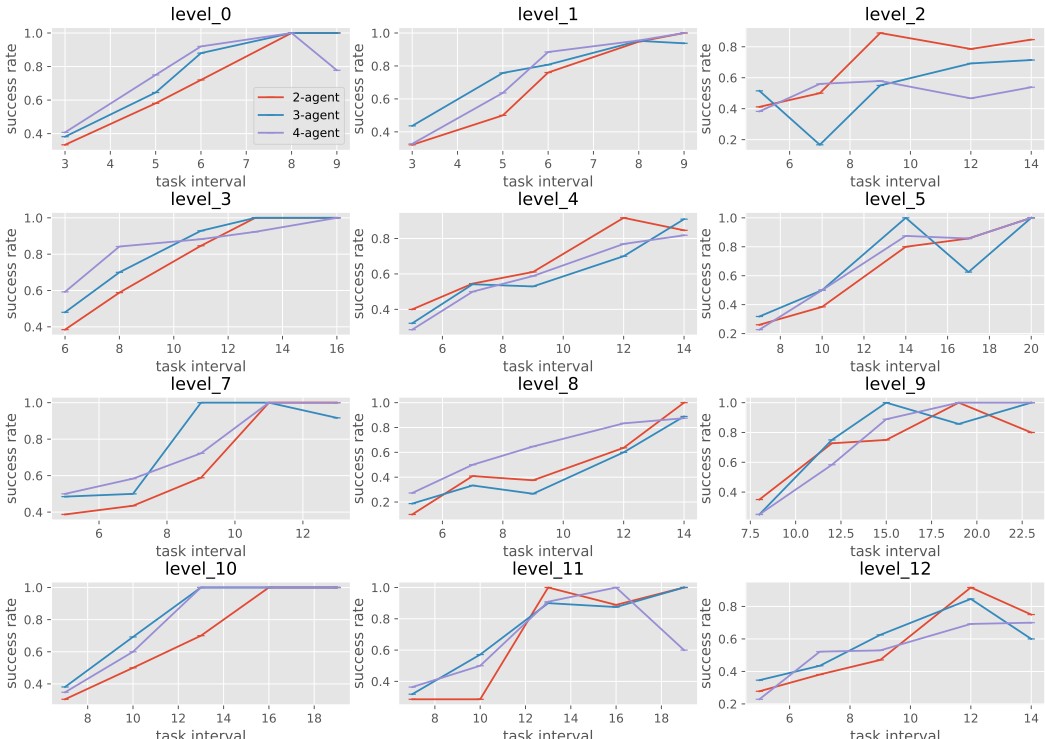

Figure 21: Collaboration results on different tasks

## E.1 COMPARISON BETWEEN CUISINEWORLD AND RELATED BENCHMARKS

| Benchmark | Multi-task | Object Interaction | Tool Use | Maximum Agents | Collabo-ration | Human in-the-loop | Procedural Level Generation |
|---|---|---|---|---|---|---|---|
| ALFWorld (Shridhar et al., 2020) | ✓ | ✓ | ✓ | 1 | ✗ | ✗ | ✗ |
| WAH (Puig et al., 2020) | ✓ | ✓ | ✗ | 2 | ✓ | ✓ | ✗ |
| TextWorld (Côté et al., 2019) | ✓ | ✓ | ✓ | 1 | ✗ | ✗ | ✓ |
| Generative Agents (Park et al., 2023) | ✓ | ✓ | ✓ | 25 | ✗ | ✗ | ✓ |
| EMATP (Liu et al., 2022) | ✓ | ✓ | ✓ | 2 | ✓ | ✗ | ✗ |
| Overcooked-AI (Carroll et al., 2019) | ✗ | ✓ | ✓ | 2 | ✓ | ✓ | ✗ |
| HandMeThat (Wan et al., 2022) | ✓ | ✓ | ✓ | 2 | ✓ | ✗ | ✗ |
| DialFRED (Gao et al., 2022) | ✓ | ✓ | ✓ | 2 | ✓* | ✗ | ✗ |
| TEACH (Padmakumar et al., 2022) | ✓ | ✓ | ✓ | 2 | ✓* | ✗ | ✗ |
| CerealBar (Suhr et al., 2019) | ✗ | ✗ | ✗ | 2 | ✓ | ✗ | ✗ |
| LIGHT (Urbanek et al., 2019) | ✓ | ✗ | ✗ | 1369 | ✗ | ✓ | ✓ |
| Diplomacy (Bakhtin et al., 2022) | ✗ | ✗ | ✗ | 7 | ✓ | ✓ | ✗ |
| CordialSync (Jain et al., 2020) | ✗ | ✓ | ✗ | 2 | ✓ | ✗ | ✗ |
| CoELA (Zhang et al., 2023b) | ✓ | ✓ | ✗ | 2 | ✓ | ✓ | ✗ |
| TooManyCooks (Wu et al., 2021) | ✓ | ✓ | ✓ | 2 | ✓ | ✓ | ✗ |
| CuisineWorld (Ours) | ✓ | ✓ | ✓ | 4+ | ✓ | ✓ | ✓ |

Table 13: Comparison between CuisineWorld and other related benchmarks. *: Notably, even though multiple agents can be present, the second agent is limited to communicating with the first agent. The second agent cannot interact with the environment in an active gaming capacity.

Table 13 compares CuisineWorld against related benchmarks along the following criteria:

• **Multi-task**: The benchmark contains multiple different tasks.

- **Object Interaction**: Agents must manipulate or engage with different items or environmental elements to achieve certain goals with irreversible actions.
- **Tool Use**: Completing tasks necessitates the use of specific tools by the agents.
- **Maximum Agents**: Denotes the upper limit of agents that can be present in any experiment.
- **Collaboration**: Many tasks mandate teamwork and collaboration between different agents.
- **Human in-the-loop**: The framework allows humans to join the game and collaborate actively with the agents.
- **Procedural Level Generation**: There is flexibility in adding new tasks, making the game dynamic and adaptable.

Compared to other benchmarks like Wu et al. (2021); Carroll et al. (2019), we have the following differences:

Enhanced Kitchen Layouts: We have diversified the layout of kitchens by incorporating a wider range of tools (both in types and quantities) and ingredients. This approach differs from Wu et al. (2021) and Carroll et al. (2019), where the emphasis is not on the variety of kitchen tools and ingredients. For example, in Carroll et al. (2019), there are only two types of ingredients and two types of tools. In Wu et al. (2021), there are two types of ingredients, and two types of tools. In comparision, there are 27 unique ingredients, and 10 tools in CuisineWorld.

Complex Task Design: Our benchmark includes a broader spectrum of recipes, varying significantly in difficulty levels. This variation is not just in terms of the number of tools and ingredients required but also in the intermediate steps involved in each recipe. We invite you to refer to Figure 23 for a detailed illustration. This aspect of task complexity, particularly in the context of high-level planning, is not extensively explored in Carroll et al. (2019) and Wu et al. (2021). In Wu et al. (2021); Carroll et al. (2019) there are very limited number of dishes, 1 and 3 respectively. However, in CuisineWorld, there are 33 unique dishes.

Multi-Dish Episodes and Collaborative Strategy Assessment: We require agents to complete multiple dishes within a single episode, with the types of dishes varying to challenge and assess the collaborative strategies of the agents. Our level design ensures that there are shared intermediate steps among the types of dishes in a single episode. The system is tasked with multiple different goals at the same time. This approach allows us to use metrics like 'Collaborative Score' (CoS) to evaluate how agents collaborate to achieve higher dish throughput. This dynamic aspect of collaboration, especially in the context of dish expiration and shared tasks, offers a new dimension to the study of multi-agent cooperation, which is distinct from the environments in Carroll et al. (2019) and Wu et al. (2021). In Carroll et al. (2019) the goal is to finish as many dishes as possible in a limited amount of time. In Wu et al. (2021), the goal is to finish one dish in the least amount of time. Both of them do not consider the density of the tasks (interval between dish orders coming to the kitchen) and its effect on coordination. As we mentioned earlier, this concpet (changing density of the tasks and measuring collaboration proficiency upon it) is at heart of CuisineWorld and our CoS metric, and it has demonstrated its effectiveness of benchmarking collaboration between LLMs and human-NPCs (as indicated in the abstract).

### E.2 Visualizing CuisineWorld

To implement CuisineWorld into a real-world game system, we built on top of Gao et al. (2020). In our game, as visually depicted in Figure 22, players are given the opportunity to engage in collaborative interactions with NPCs. This introduces a unique dynamic to the gameplay, enabling users to experience a more immersive cooperative environment. Additionally, the game's interface is versatile, providing players multiple ways to interact within the game world. They can either use a standard keyboard setup, which is conventional and likely familiar to most PC gamers, or immerse themselves even further using a Virtual Reality (VR) device. This VR functionality ensures a more tactile and realistic interaction, as players can physically move, gesture, and engage with the NPCs and other in-game elements in the 3D environment.

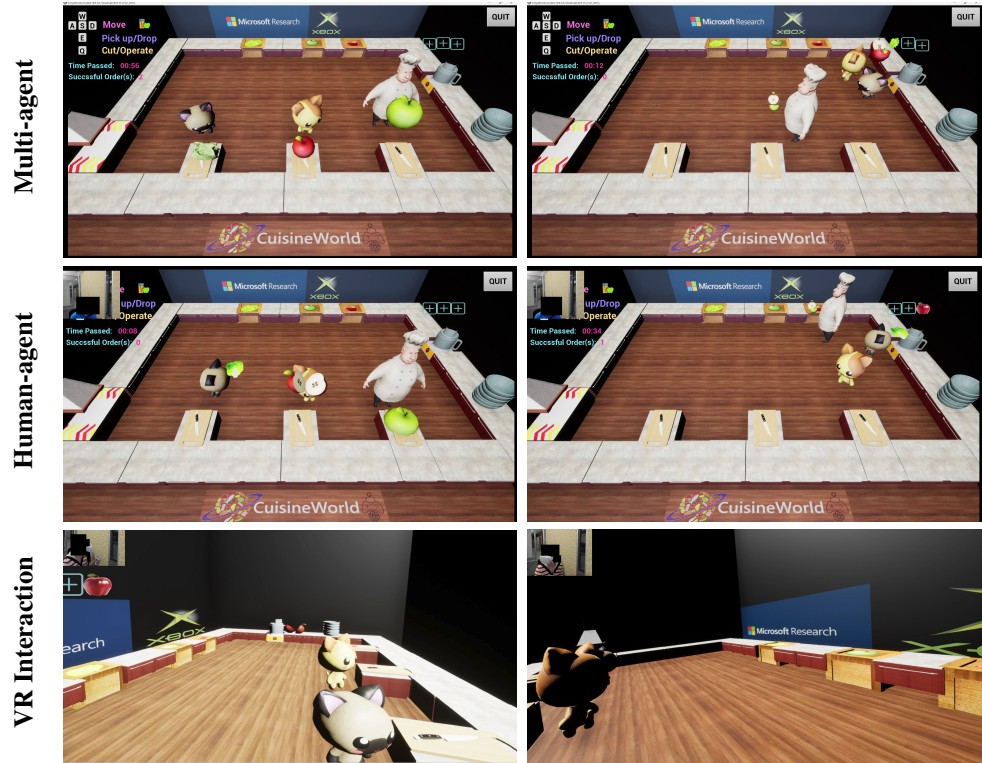

Figure 22: (Top) A multi-agent collaboration example in CuisineWorld; the three agents are preparing a mixed juice together. (Middle) A human player as the head chef instructing the agents to cook mixed juice. (Bottom) A human player collaborating with collaborative agents in VR.

Unlike the step-by-step nature of the text version, the real-time virtual game operates continuously. To align the LLM's processing with this dynamic environment, we implemented a system where the LLM checks user actions at regular intervals during the game loop, referred to as "time steps". These time steps are defined as 0.1 seconds. Then we can ensure LLM can respond to user actions in a timely manner, matching the pace of the real-time game.

In the real-time version, human players control their agents directly through keyboard inputs, resulting in low-level actions like moving or picking up items. However, the LLM-agent operates on a higher, more temporally-extended level of atomic actions. To bridge this gap, when the LLM checks user actions, it doesn't just read the keyboard inputs. Instead, it assesses changes in the high-level game state, such as the agent's location, and the status of tools and ingredients. This assessment allows the LLM to infer the temporally extended actions that align with its text-based decision-making process. Implementing this required a simple inverse dynamics model through checking state changes to translate low-level actions into high-level atomic actions, facilitating a seamless transition from the text game to the real-time virtual game.

Regarding the specific query about the 'GoTo' action, we utilized the A* pathfinding algorithm, integrated into the Unreal Engine, to facilitate this movement.

### E.3 ADDITIONAL CUISINEWORLD DETAILS

| Type | Arguments | Description |
|---|---|---|
| goto | agent location | Move agent to location |
| get | agent location (item) | agent obtain item from location |
| put | agent location | agent put everything it holds to location |
| activate | agent location | agent turn on location |
| noop | agent | not dispatching agent |

Table 14: Action space in CuisineWorld.

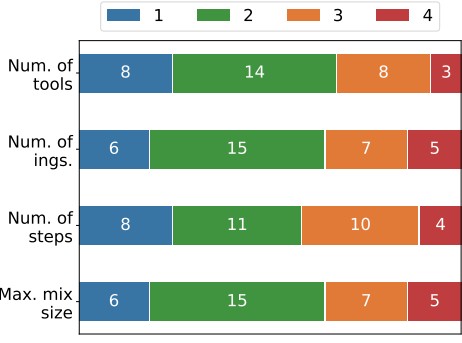

Figure 23: Dish distribution over the number of tools and ingredients (ings.) involved, cooking steps, and maximum mixture size as in the recipe.

### E.4 TASK INTERVAL COMPUTATION

In our approach, we initially construct a task graph delineating subgoals, which serves as the foundation for our computations. We then apply breadth-first search in a single-agent context to determine the optimal task sequence. This sequence is a key component for calculating task intervals in multi-agent collaboration scenarios. For each tool requiring activation, we incorporate its activation wait time into the respective task intervals. Additionally, for each new connection in the task graph, we increase the total task interval time by tripling the edge time. We assume each subgoal requires at least, goto, get and put 3 actions. The cumulative task interval is subsequently adjusted by a scaling factor of 0.3 and the variable $\tau$. We pick the value $tau$ ranging from 1.0, 1.5, 2.0, 2.5 and 3.0 to represent different task difficulties. This process enables us to effectively compute task intervals tailored for multi-agent collaborative environments of varied difficulties.

### E.5 COMMON FAILURE MODES

Through replaying actions, we have identified the following common failure modes for GPT-4 agents: 1) Inability to Prioritize Task Order: Occasionally, the LLM overlooks the task at the top of the queue, leading to the expiration of that task. 2) Difficulty Understanding the 'Occupy()' State Instruction: In CuisineWorld, agents must wait for **varying** timesteps before cooking is completed, with the wait time dependent on the specific tool used. If agents attempt to remove ingredients immediately after activating a tool, the action fails. Instead of continuing to wait, the agents may shift their focus to other tasks, which slows down overall progress. 3) Challenges in Allocating Agents to Correct Subgoals: When multiple dishes are being prepared concurrently, the agents often struggle to allocate themselves effectively to the appropriate subgoals.

### E.6 RATIONAL FOR COS METRIC

- In the kitchen scenario as demonstrated in CuisineWorld, hypothetically, when the dish order come very rarely (with a large interval), no matter if there is any collaboration, high success rate can easily attained as there is sufficient time.

- However, as we reduce the interval, more and more dish order are flooding in. If the agents (and humans) are able to collaborate well, the productivity will be high, or namely, they can still manage to maintain a decent success rate. On the contrary, teams with poor collaborate will likely suffer from a substantial drop on success rate as the interval gets smaller. The same collapse will ultimately happen to a good collaborative team too when the interval gets too small, but good collaboration can always sustain longer.

- Therefore, it makes sense to use the averaged success rate across different intervals as an indicator of the collaboration proficiency. More importantly, such metric asks for a game setting where the dish order will keep coming, in a changing interval, which also aligns with the original Overcooked! game experiences.

# F  MINECRAFT

## F.1  TRANSFER TO MINECRAFT

To transfer MindAgent from CuisineWorld to Minecraft requires modifications on both games and the model. On the LLM side, we update the background knowledge of the model. This included: 1) Action Space Explanation: We provided the LLM with detailed information about the possible actions and interactions within the Minecraft environment. 2) Recipe and Tool Definitions: We also included definitions of recipes, tools, and ingredients specific to Minecraft. We believe these modifications are reasonable and necessary, as without these knowledge, it's very difficult for model to operate in an unknown environments. On the game development side, we dedicated efforts to: Text-to-Game Interaction Translation: We developed code to translate text-based interactions and commands from the LLM into actionable inputs within the game environment. This translation layer was key to bridging the gap between the LLM's text-based outputs and the game's interactive elements.

## F.2  TASK GRAPHS

In Figure 24 we visualize the task graphs for different tasks in Minecraft.

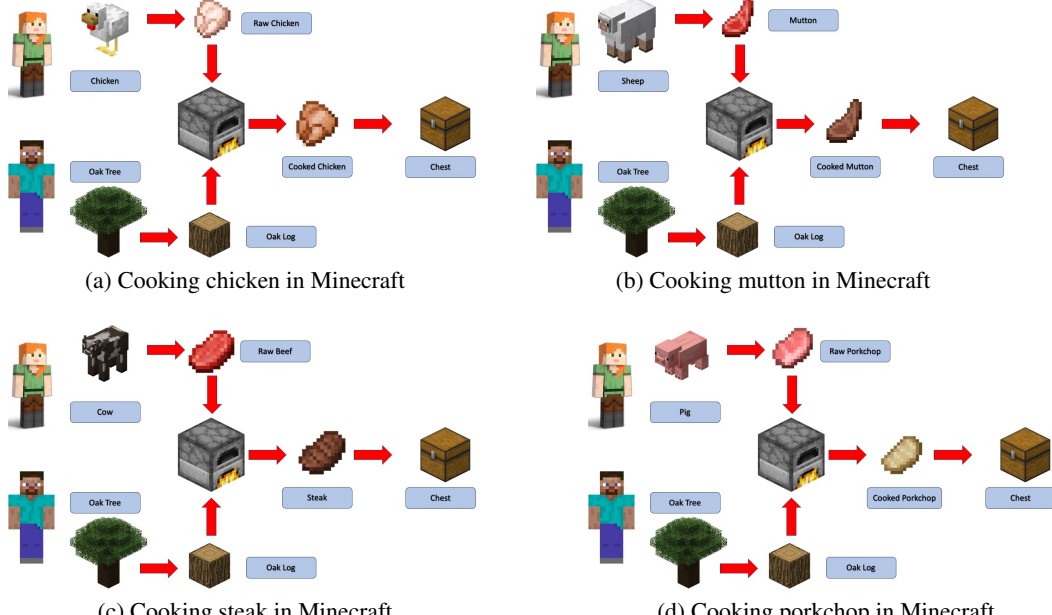

(a) Cooking chicken in Minecraft

(b) Cooking mutton in Minecraft

(c) Cooking steak in Minecraft

(d) Cooking porkchop in Minecraft

Figure 24: Task Visulization in Minecraft

## F.3 Gameplay Visualization

We visualize Minecraft gameplay in Figure 25.

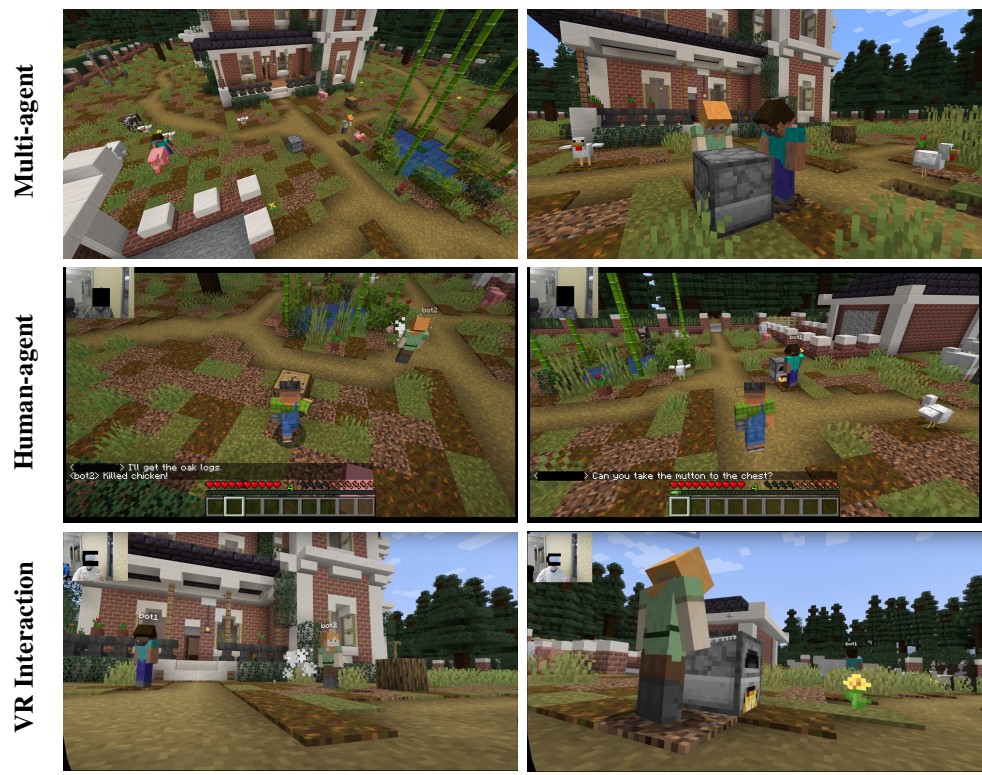

Figure 25: (Top) A multi-agent collaboration example in Minecraft. At left Alex and Steve are killing different animals and at right they are cooking meat in a furnace together. (Middle) A human player instructing the agents to perform certain actions. (Bottom) A human player collaborating with agents in VR.

## F.4 Action Details for Mindcraft

We define the following actions for the multi-agent system in our Minecraft game: 1) `goto(agent, location)`; 2) `killMob(agent, mobType)`; 3) `mineBlock(agent, blockType)`; 4) `putFuelFurnace(agent, fuelType)`, to put the item from agent's inventory to the furnace's bottom slot. 5) `putItemFurnace(agent, itemType)`, to put the item from agent's inventory to the furnace's top slot; 6) `takeOutFurnace(agent)`, take out the cooked item from the furnace 7) `putInChest(agent, itemType)`.

The state space in Minecraft contains the following: 1) nearby blocks for each agent, 2) nearby entities for each agent, 3) each agent's inventory, 4) items inside the furnace, 5) items inside the chest, and 6) the human player's inventory if a human player is involved.

To ensure reproducibility, we modify the game mechanism. A killed mob will respawn nearby, and a mined block will also respawn nearby.

# G ADDITIONAL INFORMATION ON HUMAN EVALUATION

## G.0.1 HUMAN DATA COLLECTION

**Measurement.** In the background, we collect the numbers of failed and successful tasks during a participant's interaction with the game system. Additionally, we record the entire action history of players and intelligent agents. After each episode, the participants must complete a survey about their engagement with the system on a 5-point Likert chart. Our objective measure is intended to evaluate the human-AI teaming performance, and the subjective measure is designed to evaluate users' perceptions of the system. The human evaluation interface can be found in Appendix G.

## G.1 HUMAN EVALUATION INTERFACE

We use the human evaluation interface to test the human's perception of collaborative agents. This gives us a more controlled environment so users' perception of collaborative agents does not depend on their ability to control the keyboard and mouse, and their perception of collaborative agents does not depend on the latency and rate limits of GPT-4. Figure 26 shows the interface welcome screen, human evaluation examples, and examples of human instructions.

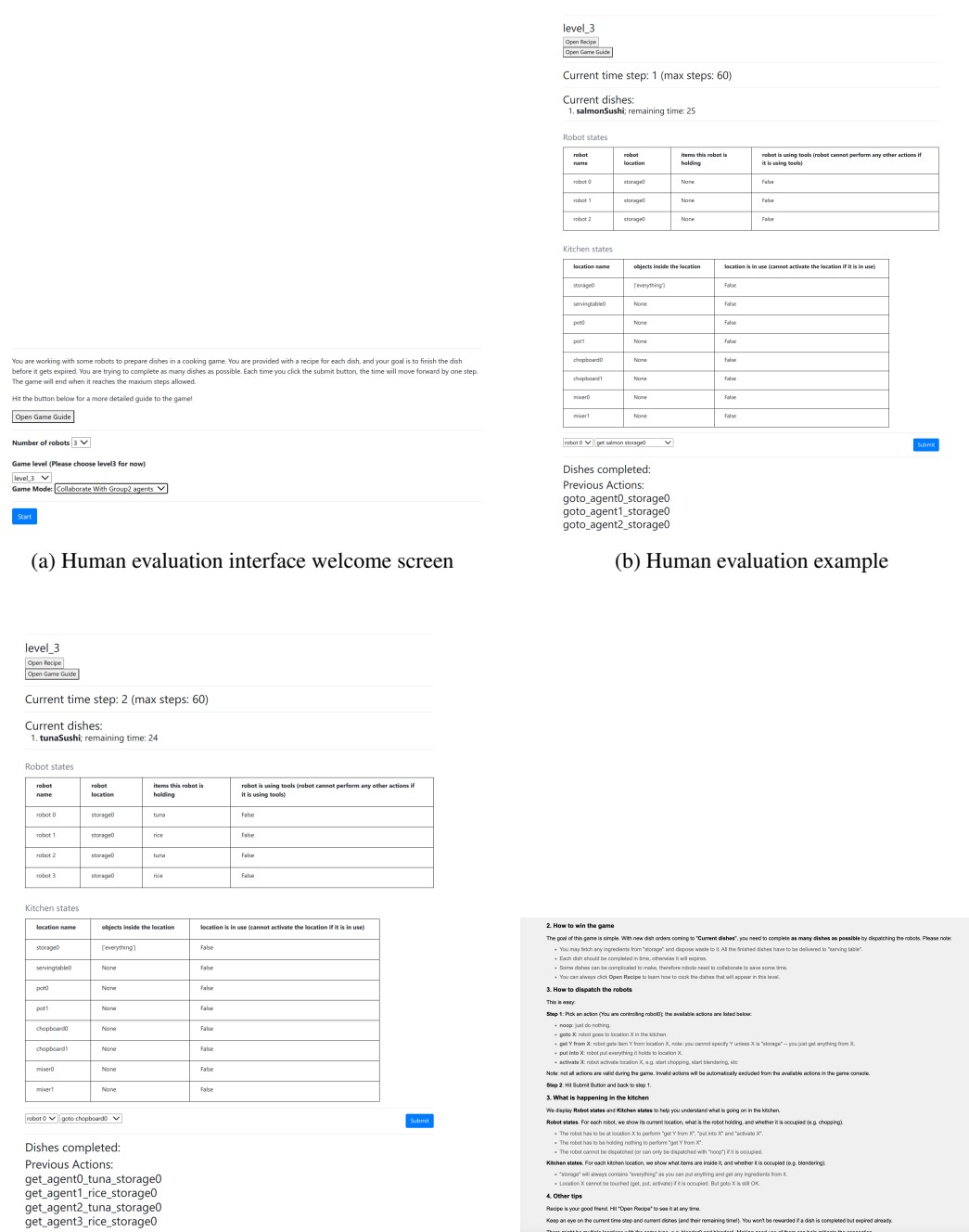

(a) Human evaluation interface welcome screen

(b) Human evaluation example

(c) Human evaluation example

(d) Human instructions

Figure 26: Human evaluation interface welcome screen (a), evaluation examples (b)–(c), and instructions to the human participants (d).

## G.2   HUMAN EVALUATION QUESTIONNAIRE

## G.3 Additional Results on Human Evaluation

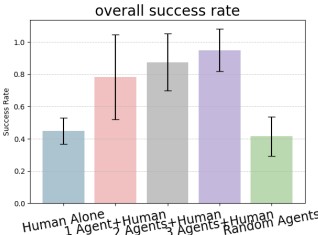
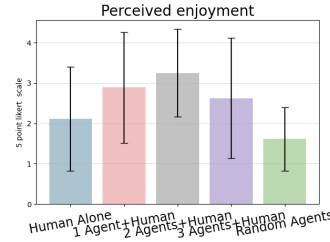
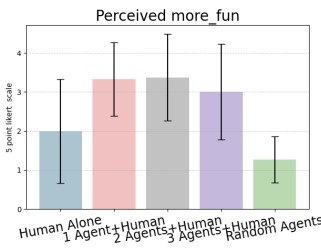

(a) **Collaboration score:**
The collaboration score is higher if more agents are collaborating with human players, although the difference is not significant.

(b) **Perceived enjoyment:**
Humans enjoy the game more if they collaborate with the right number of agents.

(c) **Perceived more fun:**
Players enjoy the game more because of collaboration with competent agents.

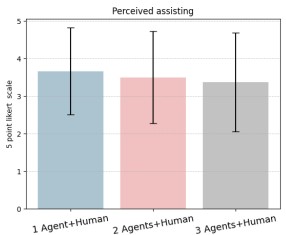
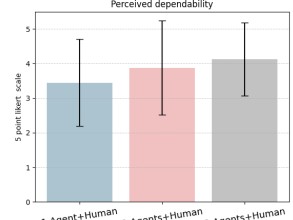
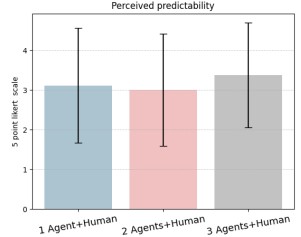

(d) **Perceived Assisting:**
There is no significant difference in terms of human perceptions of helpfulness when collaborating with more agents, even though the task success rate is higher.

(e) **Perceived dependability:**
When collaborating with more agents, players depend on the agents more.

(f) **Perceived Predictability:**
There is no difference in terms of the predictability of agent behaviors when collaborating with more agents.

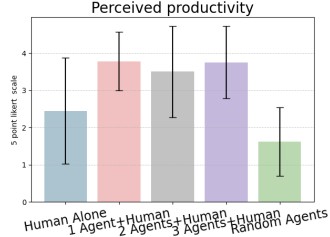
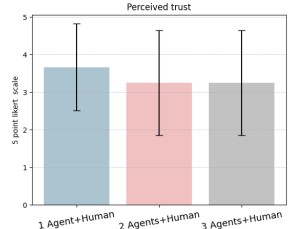

(g) **Perceived productivity:**
Players think collaborating with AI agents will improve productivity.

(h) **Perceived Trust:**
There is no difference in terms of trust when collaborating with more agents.

Figure 27: Full results of human evaluations

## H Full Interaction History

The full interaction history for finishing one dish in level 9 showed as the below.

```
The available actions are :
1) goto: goto a tool location
2) get: get some object from a tool
3) put: put some object into a tool
4) activate: activate the tool to cook all ingredients
inside the tool into a different tools
5) noop: not performing any actions
Sometimes the system will give you error messages.
Please consider these error messages when executing actions.
```

You need to specify action for all of the agents, **except human**.
They all have different agent numbers.
Do not assign actions to the same agent more than once.

When the tools reach its capacity, you need to take stuff out.
Otherwise, you cannot put items inside.
When you are holding objects, you cannot get any more objects.
When you are holding objects, you cannot activate tools.
Afer you cooked a required dish, you need to put it into the servingtable.
You can only pick up objects from the tool location,
if you are located at the tool location.
When you activate any tools,
make sure all the items inside the tool are respecting the recipes.
Otherwise, you will cook waste. Avoid waste at all cost.
*** You should mix salad in the mixer.
To make salad you should chop veggies first. ***
*** If the tool is occupied, indicated by the occupy() predicate,
you cannot get objects from it or put objects into it. ***
*** The food orders are keep coming.
You should finish as many dishes as possible
and finish every dish as soon as possible.
Please deliver the order to the serveringtable when it is finished. ***
*** The dish will expire after the lifetime reaches 0
and it's not at the serveringtable.
Please avoid this. *** Here are the recipes:

Cook porkMeatcake at:
 -- location: blender
 -- with ingredients:        pork,        flour,
Cook lettuceSlice at:
 -- location: chopboard
 -- with ingredients:        lettuce,
Cook tomatoSlice at:
 -- location: chopboard
 -- with ingredients:        tomato,
Cook sauteedBeef at:
 -- location: pan
 -- with ingredients:        beef,
Cook cheeseBurger at:
 -- location: mixer
 -- with ingredients:        cheese,        sauteedBeef,        bread,
Cook MaxJr at:
 -- location: mixer
 -- with ingredients:        lettuceSlice,        sauteedBeef,
cheese,        bread,
Cook Hopper at:
 -- location: mixer
 -- with ingredients:        tomatoSlice,        lettuceSlice,
sauteedBeef,        bread,
The following objects are available:
 --1) cheese
 --2) lettuce
 --3) lettuceSlice
 --4) Hopper
 --5) MaxJr
 --6) tomato
 --7) sauteedBeef
 --8) pork
 --9) porkMeatcake

--10) cheeseBurger
--11) flour
--12) tomatoSlice
--13) bread
--14) beef
The objecsts are cooked using tools or are just base ingredients.
Among them, the following are base ingredients:
--1) cheese
--2) lettuce
--3) tomato
--4) pork
--5) flour
--6) bread
--7) beef
You can only obtain base ingredients from the storage initially.
Additional rules:
You can place up to infinite item into the storage0
You can place up to 5 item into the pan0
You can place up to 5 item into the pan1
You can place up to 5 item into the mixer0
You can place up to 5 item into the mixer1
You can place up to 1 item into the chopboard0
You can place up to 1 item into the chopboard1
You can place up to infinite item into the servingtable0
** Only ** the following tools are available:
storage0, pan0, pan1, mixer0, mixer1, chopboard0, chopboard1, servingtable0,
You cannot pick up these tools.
You can only use those tools at the corresponding location.

There are 2 agents available, so you can execute 2 actions at a time.
This is an example you can use as a reference for a different level.
*** You must follow the format below for actions

-game state:
current game level: level_1
current dishes:
    name: porkMeatcake lifetime: 17
current game step: 0
maximum game steps: 30

-agent state:
at(agent0, servingtable0)
hold(agent0, None)
at(agent1, servingtable0)
hold(agent1, None)

-kitchen state:
inside(storage0, None)
inside(servingtable0, None)
inside(blender0, None)

-accomplished task:

-action:

goto_agent0_storage0
goto_agent1_storage0

```
−game state:
current game level: level_1
current dishes:
    name: porkMeatcake lifetime: 16
current game step: 1
maximum game steps: 30

−agent state:
at(agent0, storage0)
hold(agent0, None)
at(agent1, storage0)
hold(agent1, None)

−kitchen state:
inside(storage0, None)
inside(servingtable0, None)
inside(blender0, None)

−accomplished task:

−action:

get_agent0_flour_storage0
get_agent1_pork_storage0

−game state:
current game level: level_1
current dishes:
    name: porkMeatcake lifetime: 15
current game step: 2
maximum game steps: 30

−agent state:
at(agent0, storage0)
hold(agent0, flour)
at(agent1, storage0)
hold(agent1, pork)

−kitchen state:
inside(storage0, None)
inside(servingtable0, None)
inside(blender0, None)

−accomplished task:

−action:

goto_agent0_blender0
goto_agent1_blender0

−game state:
current game level: level_1
current dishes:
    name: porkMeatcake lifetime: 14
current game step: 3
maximum game steps: 30

−agent state:
```

```
at(agent0, blender0)
hold(agent0, flour)
at(agent1, blender0)
hold(agent1, pork)

-kitchen state:
inside(storage0, None)
inside(servingtable0, None)
inside(blender0, None)

-accomplished task:

-action:

put_agent0_blender0
put_agent1_blender0

-game state:
current game level: level_1
current dishes:
    name: porkMeatcake lifetime: 13
current game step: 4
maximum game steps: 30

-agent state:
at(agent0, blender0)
hold(agent0, None)
at(agent1, blender0)
hold(agent1, None)

-kitchen state:
inside(storage0, None)
inside(servingtable0, None)
inside(blender0, flour&pork)

-accomplished task:

-action:

goto_agent0_storage0
activate_agent1_blender0

-game state:
current game level: level_1
current dishes:
    name: porkMeatcake lifetime: 12
current game step: 5
maximum game steps: 30

-agent state:
at(agent0, storage0)
hold(agent0, None)
at(agent1, blender0)
hold(agent1, None)

-kitchen state:
inside(storage0, None)
inside(servingtable0, None)
inside(blender0, porkMeatcake)
```

```
occupy ( blender0 )

−accomplished task :

−action :

get_agent0_flour_storage0
noop_agent1

−game state :
current game level : level_1
current dishes :
    name : porkMeatcake lifetime : 11
current game step : 6
maximum game steps : 30

−agent state :
at ( agent0 , storage0 )
hold ( agent0 , flour )
at ( agent1 , blender0 )
hold ( agent1 , None )

−kitchen state :
inside ( storage0 , None )
inside ( servingtable0 , None )
inside ( blender0 , porkMeatcake )

−accomplished task :

−action :

goto_agent0_blender0
get_agent1_blender0

−game state :
current game level : level_1
current dishes :
    name : porkMeatcake lifetime : 10
    name : porkMeatcake lifetime : 17
current game step : 7
maximum game steps : 30

−agent state :
at ( agent0 , blender0 )
hold ( agent0 , flour )
at ( agent1 , blender0 )
hold ( agent1 , porkMeatcake )

−kitchen state :
inside ( storage0 , None )
inside ( servingtable0 , None )
inside ( blender0 , None )

−accomplished task :

−action :

put_agent0_blender0
goto_agent1_servingtable0
```

```
−game state :
current game level : level_1
current dishes :
    name: porkMeatcake lifetime : 9
    name: porkMeatcake lifetime : 16
current game step : 8
maximum game steps : 30

−agent state :
at(agent0 , blender0 )
hold(agent0 , None)
at(agent1 , servingtable0 )
hold(agent1 , porkMeatcake )

−kitchen state :
inside(storage0 , None)
inside(servingtable0 , None)
inside(blender0 , flour )

−accomplished task :

−action :

goto_agent0_storage0
put_agent1_servingtable0

−game state :
current game level : level_1
current dishes :
    name: porkMeatcake lifetime : 15
current game step : 9
maximum game steps : 30

−agent state :
at(agent0 , storage0 )
hold(agent0 , None)
at(agent1 , servingtable0 )
hold(agent1 , None)

−kitchen state :
inside(storage0 , None)
inside(servingtable0 , None)
inside(blender0 , flour )

−accomplished task :
porkMeatcake ,

−action :

get_agent0_pork_storage0
goto_agent1_blender0

In this game , There are 2 agents available and no human ,
so you should control all the 2 agents , and plan 2 actions at a time .

[[ user ]]
```

```
-execution error messages:
  -- []
-execution suggestions:
  -- []
-game state:
current game level: level_9
current dishes:
    name: MaxJr lifetime: 17
current game step: 0
maximum game steps: 60

-agent state:
at(agent0, servingtable0)
hold(agent0, None)
at(agent1, servingtable0)
hold(agent1, None)

-kitchen state:
inside(storage0, None)
inside(pan0, None)
inside(pan1, None)
inside(mixer0, None)
inside(mixer1, None)
inside(chopboard0, None)
inside(chopboard1, None)
inside(servingtable0, None)

-accomplished task:

-action:

[[assistant]]

goto_agent0_storage0
goto_agent1_storage0

[[user]]

-execution error messages:
  -- []
-execution suggestions:
  -- []
-game state:
current game level: level_9
current dishes:
    name: MaxJr lifetime: 16
current game step: 1
maximum game steps: 60

-agent state:
at(agent0, storage0)
hold(agent0, None)
at(agent1, storage0)
hold(agent1, None)

-kitchen state:
inside(storage0, None)
```

```
inside(pan0, None)
inside(pan1, None)
inside(mixer0, None)
inside(mixer1, None)
inside(chopboard0, None)
inside(chopboard1, None)
inside(servingtable0, None)

-accomplished task:

-action:

[[assistant]]

get_agent0_lettuce_storage0
get_agent1_beef_storage0

[[user]]

-execution error messages:
    --  []
-execution suggestions:
    --  []
-game state:
current game level: level_9
current dishes:
    name: MaxJr lifetime: 15
current game step: 2
maximum game steps: 60

-agent state:
at(agent0, storage0)
hold(agent0, lettuce)
at(agent1, storage0)
hold(agent1, beef)

-kitchen state:
inside(storage0, None)
inside(pan0, None)
inside(pan1, None)
inside(mixer0, None)
inside(mixer1, None)
inside(chopboard0, None)
inside(chopboard1, None)
inside(servingtable0, None)

-accomplished task:

-action:

[[assistant]]

goto_agent0_chopboard0
goto_agent1_pan0

[[user]]
```

```
-execution error messages:
   --  []
-execution suggestions:
   --  []
-game state:
current game level: level_9
current dishes:
    name: MaxJr lifetime: 14
current game step: 3
maximum game steps: 60

-agent state:
at(agent0, chopboard0)
hold(agent0, lettuce)
at(agent1, pan0)
hold(agent1, beef)

-kitchen state:
inside(storage0, None)
inside(pan0, None)
inside(pan1, None)
inside(mixer0, None)
inside(mixer1, None)
inside(chopboard0, None)
inside(chopboard1, None)
inside(servingtable0, None)

-accomplished task:

-action:

[[assistant]]

put_agent0_chopboard0
put_agent1_pan0

[[user]]

-execution error messages:
   --  []
-execution suggestions:
   --  []
-game state:
current game level: level_9
current dishes:
    name: MaxJr lifetime: 13
current game step: 4
maximum game steps: 60

-agent state:
at(agent0, chopboard0)
hold(agent0, None)
at(agent1, pan0)
hold(agent1, None)

-kitchen state:
inside(storage0, None)
inside(pan0, beef)
```

```
inside(pan1, None)
inside(mixer0, None)
inside(mixer1, None)
inside(chopboard0, lettuce)
inside(chopboard1, None)
inside(servingtable0, None)

-accomplished task:

-action:

[[assistant]]

activate_agent0_chopboard0
activate_agent1_pan0

[[user]]

-execution error messages:
    -- []
-execution suggestions:
    -- []
-game state:
current game level: level_9
current dishes:
    name: MaxJr lifetime: 12
current game step: 5
maximum game steps: 60

-agent state:
at(agent0, chopboard0)
hold(agent0, None)
at(agent1, pan0)
hold(agent1, None)

-kitchen state:
inside(storage0, None)
inside(pan0, sauteedBeef)
occupy(pan0)
inside(pan1, None)
inside(mixer0, None)
inside(mixer1, None)
inside(chopboard0, lettuceSlice)
inside(chopboard1, None)
inside(servingtable0, None)

-accomplished task:

-action:

[[assistant]]

goto_agent0_storage0
get_agent1_pan0

[[user]]
```

```
-execution error messages:
   -- ['pan0 is occupied, therefore agent1 cannot get objects from the tool']
-execution suggestions:
   -- []
-game state:
current game level: level_9
current dishes:
    name: MaxJr lifetime: 11
current game step: 6
maximum game steps: 60

-agent state:
at(agent0, storage0)
hold(agent0, None)
at(agent1, pan0)
hold(agent1, None)

-kitchen state:
inside(storage0, None)
inside(pan0, sauteedBeef)
inside(pan1, None)
inside(mixer0, None)
inside(mixer1, None)
inside(chopboard0, lettuceSlice)
inside(chopboard1, None)
inside(servingtable0, None)

-accomplished task:

-action:

[[assistant]]

get_agent0_cheese_storage0
noop_agent1

[[user]]

-execution error messages:
   -- []
-execution suggestions:
   -- []
-game state:
current game level: level_9
current dishes:
    name: MaxJr lifetime: 10
current game step: 7
maximum game steps: 60

-agent state:
at(agent0, storage0)
hold(agent0, cheese)
at(agent1, pan0)
hold(agent1, None)

-kitchen state:
inside(storage0, None)
inside(pan0, sauteedBeef)
```

```
inside(pan1, None)
inside(mixer0, None)
inside(mixer1, None)
inside(chopboard0, lettuceSlice)
inside(chopboard1, None)
inside(servingtable0, None)

-accomplished task:

-action:

[[assistant]]

goto_agent0_mixer0
get_agent1_pan0

[[user]]

-execution error messages:
   -- []
-execution suggestions:
   -- []
-game state:
current game level: level_9
current dishes:
    name: MaxJr lifetime: 9
    name: MaxJr lifetime: 17
current game step: 8
maximum game steps: 60

-agent state:
at(agent0, mixer0)
hold(agent0, cheese)
at(agent1, pan0)
hold(agent1, sauteedBeef)

-kitchen state:
inside(storage0, None)
inside(pan0, None)
inside(pan1, None)
inside(mixer0, None)
inside(mixer1, None)
inside(chopboard0, lettuceSlice)
inside(chopboard1, None)
inside(servingtable0, None)

-accomplished task:

-action:

[[assistant]]

put_agent0_mixer0
goto_agent1_chopboard0

[[user]]
```

```
−execution error messages:
  −−  []
−execution suggestions:
  −−  []
−game state:
current game level: level_9
current dishes:
    name: MaxJr lifetime: 8
    name: MaxJr lifetime: 16
current game step: 9
maximum game steps: 60

−agent state:
at(agent0, mixer0)
hold(agent0, None)
at(agent1, chopboard0)
hold(agent1, sauteedBeef)

−kitchen state:
inside(storage0, None)
inside(pan0, None)
inside(pan1, None)
inside(mixer0, cheese)
inside(mixer1, None)
inside(chopboard0, lettuceSlice)
inside(chopboard1, None)
inside(servingtable0, None)

−accomplished task:

−action:

[[assistant]]

goto_agent0_storage0
get_agent1_chopboard0

[[user]]

−execution error messages:
  −−  ['agent1 is holding objects, therefore cannot get objects from the tool']
−execution suggestions:
  −−  []
−game state:
current game level: level_9
current dishes:
    name: MaxJr lifetime: 7
    name: MaxJr lifetime: 15
current game step: 10
maximum game steps: 60

−agent state:
at(agent0, storage0)
hold(agent0, None)
at(agent1, chopboard0)
hold(agent1, sauteedBeef)

−kitchen state:
```

```
inside(storage0, None)
inside(pan0, None)
inside(pan1, None)
inside(mixer0, cheese)
inside(mixer1, None)
inside(chopboard0, lettuceSlice)
inside(chopboard1, None)
inside(servingtable0, None)

-accomplished task:

-action:

[[assistant]]

get_agent0_bread_storage0
put_agent1_mixer0

[[user]]

-execution error messages:
    -- ['agent1 is not located in mixer0']
-execution suggestions:
    -- []
-game state:
current game level: level_9
current dishes:
    name: MaxJr lifetime: 6
    name: MaxJr lifetime: 14
current game step: 11
maximum game steps: 60

-agent state:
at(agent0, storage0)
hold(agent0, bread)
at(agent1, chopboard0)
hold(agent1, sauteedBeef)

-kitchen state:
inside(storage0, None)
inside(pan0, None)
inside(pan1, None)
inside(mixer0, cheese)
inside(mixer1, None)
inside(chopboard0, lettuceSlice)
inside(chopboard1, None)
inside(servingtable0, None)

-accomplished task:

-action:

[[assistant]]

goto_agent0_mixer0
goto_agent1_mixer0
```

```
[[user]]

-execution error messages:
  --  []
-execution suggestions:
  --  []
-game state:
current game level: level_9
current dishes:
    name: MaxJr lifetime: 5
    name: MaxJr lifetime: 13
current game step: 12
maximum game steps: 60

-agent state:
at(agent0, mixer0)
hold(agent0, bread)
at(agent1, mixer0)
hold(agent1, sauteedBeef)

-kitchen state:
inside(storage0, None)
inside(pan0, None)
inside(pan1, None)
inside(mixer0, cheese)
inside(mixer1, None)
inside(chopboard0, lettuceSlice)
inside(chopboard1, None)
inside(servingtable0, None)

-accomplished task:

-action:

[[assistant]]

put_agent0_mixer0
put_agent1_mixer0

[[user]]

-execution error messages:
  --  []
-execution suggestions:
  --  []
-game state:
current game level: level_9
current dishes:
    name: MaxJr lifetime: 4
    name: MaxJr lifetime: 12
current game step: 13
maximum game steps: 60

-agent state:
at(agent0, mixer0)
hold(agent0, None)
at(agent1, mixer0)
hold(agent1, None)
```

```
-kitchen state:
inside(storage0, None)
inside(pan0, None)
inside(pan1, None)
inside(mixer0, cheese&bread&sauteedBeef)
inside(mixer1, None)
inside(chopboard0, lettuceSlice)
inside(chopboard1, None)
inside(servingtable0, None)

-accomplished task:

-action:

[[assistant]]

activate_agent0_mixer0
goto_agent1_chopboard0

[[user]]

-execution error messages:
   -- []
-execution suggestions:
   -- []
-game state:
current game level: level_9
current dishes:
    name: MaxJr lifetime: 3
    name: MaxJr lifetime: 11
current game step: 14
maximum game steps: 60

-agent state:
at(agent0, mixer0)
hold(agent0, None)
at(agent1, chopboard0)
hold(agent1, None)

-kitchen state:
inside(storage0, None)
inside(pan0, None)
inside(pan1, None)
inside(mixer0, cheeseBurger)
inside(mixer1, None)
inside(chopboard0, lettuceSlice)
inside(chopboard1, None)
inside(servingtable0, None)

-accomplished task:

-action:

[[assistant]]

get_agent0_mixer0
```

```
get_agent1_chopboard0

[[ user ]]

-execution error messages:
  -- []
-execution suggestions:
  -- []
-game state:
current game level: level_9
current dishes:
    name: MaxJr lifetime: 2
    name: MaxJr lifetime: 10
current game step: 15
maximum game steps: 60

-agent state:
at(agent0, mixer0)
hold(agent0, cheeseBurger)
at(agent1, chopboard0)
hold(agent1, lettuceSlice)

-kitchen state:
inside(storage0, None)
inside(pan0, None)
inside(pan1, None)
inside(mixer0, None)
inside(mixer1, None)
inside(chopboard0, None)
inside(chopboard1, None)
inside(servingtable0, None)

-accomplished task:

-action:

[[ assistant ]]

goto_agent0_mixer0
put_agent1_mixer0

[[ user ]]

-execution error messages:
  -- ['agent1 is not located in mixer0']
-execution suggestions:
  -- []
-game state:
current game level: level_9
current dishes:
    name: MaxJr lifetime: 1
    name: MaxJr lifetime: 9
    name: cheeseBurger lifetime: 12
current game step: 16
maximum game steps: 60

-agent state:
at(agent0, mixer0)
```

```
hold(agent0, cheeseBurger)
at(agent1, chopboard0)
hold(agent1, lettuceSlice)

-kitchen state:
inside(storage0, None)
inside(pan0, None)
inside(pan1, None)
inside(mixer0, None)
inside(mixer1, None)
inside(chopboard0, None)
inside(chopboard1, None)
inside(servingtable0, None)

-accomplished task:

-action:

[[assistant]]

put_agent0_mixer0
goto_agent1_mixer0

[[user]]

-execution error messages:
  -- ['putting cheeseBurger into mixer0 will result in waste']
-execution suggestions:
  -- []
-game state:
current game level: level_9
current dishes:
    name: MaxJr lifetime: 0
    name: MaxJr lifetime: 8
    name: cheeseBurger lifetime: 11
current game step: 17
maximum game steps: 60

-agent state:
at(agent0, mixer0)
hold(agent0, cheeseBurger)
at(agent1, mixer0)
hold(agent1, lettuceSlice)

-kitchen state:
inside(storage0, None)
inside(pan0, None)
inside(pan1, None)
inside(mixer0, None)
inside(mixer1, None)
inside(chopboard0, None)
inside(chopboard1, None)
inside(servingtable0, None)

-accomplished task:

-action:
```

[[ assistant ]]

goto_agent0_servingtable0
put_agent1_mixer0

[[ user ]]

-execution error messages:
  -- []
-execution suggestions:
  -- []
-game state:
current game level: level_9
current dishes:
    name: MaxJr lifetime: 7
    name: cheeseBurger lifetime: 10
current game step: 18
maximum game steps: 60

-agent state:
at(agent0, servingtable0)
hold(agent0, cheeseBurger)
at(agent1, mixer0)
hold(agent1, None)

-kitchen state:
inside(storage0, None)
inside(pan0, None)
inside(pan1, None)
inside(mixer0, lettuceSlice)
inside(mixer1, None)
inside(chopboard0, None)
inside(chopboard1, None)
inside(servingtable0, None)

-accomplished task:

-action:

[[ assistant ]]

put_agent0_servingtable0
goto_agent1_storage0

[[ user ]]

-execution error messages:
  -- []
-execution suggestions:
  -- []
-game state:
current game level: level_9
current dishes:
    name: MaxJr lifetime: 6
current game step: 19
maximum game steps: 60

```
−agent state :
at ( agent0 , servingtable0 )
hold ( agent0 , None )
at ( agent1 , storage0 )
hold ( agent1 , None )

−kitchen state :
inside ( storage0 , None )
inside ( pan0 , None )
inside ( pan1 , None )
inside ( mixer0 , lettuceSlice )
inside ( mixer1 , None )
inside ( chopboard0 , None )
inside ( chopboard1 , None )
inside ( servingtable0 , None )

−accomplished task :
cheeseBurger ,
```

