# OpenReview forum: "MindAgent: Emergent Gaming Interaction"
_ICLR.cc/2024/Conference — ICLR 2024 Conference Withdrawn Submission_

### Official Review · Reviewer_YZDa · 2023-10-29

**Soundness:** 1 poor
**Presentation:** 3 good
**Contribution:** 2 fair
**Rating:** 3
**Confidence:** 5

**Summary:**

The paper introduces a new text-based gaming multi-agent benchmark CuisineWorld, inspired by the video game Overcooked, proposes a method MindAgent for central control of multiple agents using LLM by designing prompts, and conducts experiments with the benchmark and the method including human experiments.

**Strengths:**

- The paper introduces a new text-based multi-agent cooperation benchmark, with a vivid visual appearance.

- The paper conducts experiments involving more than 2 agents and human.

- The paper shows some efforts in transferring the method in real-world gaming scenarios Mindcraft.

**Weaknesses:**

1. The Benchmark's Contribution Falls Short

- The CuisineWorld, based on the video game Overcooked which is popular for measuring multi-agent cooperation and has many existing environments already[1][2]. The benchmark introduces more cuisines but falls short in diversifying kitchen maps when compared to previous Overcooked environments.

- Table 12 made a good summary of related benchmarks though some related embodied multi-agent cooperation benchmarks with most of the features in the table are missing, such as [3][4].

- While CuisineWorld boasts an appealing visual design, it does not directly align with the paper's experiments. The paper primarily relies on text-based states and interactions (Figure 26, even during human experiments, where humans are restricted to a textual interface with lots of text description of the game state)

- In section D.2, the statement "In this game, the actions of human players can be obtained from an inverse dynamic model by checking pre-conditions and post-effects" is confusing to me. How was Figure 22 obtained? It seems to be in real-time with human players using the keyboard to move, how's the time step defined here and the "goto" action for the LLM agent implemented?

- The benchmark predominantly employs a "new auto-metric collaboration score CoS for assessing the collaboration efficiency". However, this metric is defined as the average task success rate with different time intervals for each dish that appears highly tailored to the specific environment and lacks a clear connection to "cooperation efficiency".

- The absence of a train/test split is concerning, as prompt engineering can substantially impact performance. A thorough understanding of how the prompt is tuned for different tasks is crucial.

- From C.3.4, level 3 has only two similar recipes which differ only with the words "salmon" and "tuna", only one demonstration may decrease the task difficulty significantly for the LMs, raising doubts about the benchmark formation.

2. Claims Need Stronger Support from Results

- In section 5.1, The paper claims "more agents will lead to higher collaboration efficiencies. Thus, indicating that the LLM dispatcher can coordinate additional agents to execute tasks more efficiently". However, from Table 1, 4 agents perform worse than 3 agents in most scenarios, and even 2 agents achieve the highest score in levels 2 and 4, which contradicts the claim. It may provide more insights if the paper can provide more analysis on these contradictory results instead of only ablating on level_3, which seems to be the only level that "looks normal" (a.k.a 2 agents < 3 agents < 4 agents with somewhat clear gaps)

- Table 4, "For a fair comparison, all tests employed identical prompt inputs" using "identical" prompt for different LM families may not be "fair", especially if the prompt is "tuned" specifically for one model. More details on the prompt engineering process may help clarify these concerns.

- Table 3 presents a perplexing scenario where four agents using a two-agent demo outperform four agents using a four-agent demo, albeit marginally. This result could benefit from a clearer explanation.

- Novel game adaptation of Minecraft seems very promising, but the details are extremely limited. It would be more convincing if there were more details and formal experiments on it. For example, how is the "adaptation" conducted? What's the additional human effort required (such as prompt engineering and rounds of re-playing for prompt tuning)?

3. Method's Limited Contribution

- The method employed in the paper primarily relies on intensive prompt engineering, without introducing novel designs for multi-agent cooperation.

- It's well known that LLMs can perform "in-context learning", providing demonstrations and reasoning steps can help improve performance, there's nothing new to take away from the method and experiments.

- As mentioned above, the "emergent ability" of MindAgent is not well supported by the results.

- Providing only screenshots of part of the prompts as in Figures 6 and 7 is not enough. More details on the full prompt and game episode may help clarify these concerns.

[1] On the utility of learning about humans for human-ai coordination
[2] Too many cooks: Bayesian inference for coordinating multi-agent collaboration
[3] Building cooperative embodied agents modularly with large language models
[4] A Cordial Sync: Going Beyond Marginal Policies for Multi-Agent Embodied Tasks

**Questions:**

Please address the concerns raised in the Weakness.

---

> ### Author Response · Authors · 2023-11-21
> **Response to Reviewer YZDa 1/N**
>
> > The CuisineWorld, based on the video game Overcooked which is popular for measuring multi-agent cooperation and has many existing environments already[1][2]. The benchmark introduces more cuisines but falls short in diversifying kitchen maps when compared to previous Overcooked environments.
>
> We are grateful for your insightful comment regarding the design of our CuisineWorld benchmark in relation to existing environments based on the popular video game Overcooked. Our approach, indeed, differs from the environments presented in references [1] and [2]. While these existing environments offer a variety of kitchen maps, our benchmark is specifically tailored to explore high-level multi-agent planning and coordination using Large Language Model-based Multi-agent Systems (MAS).
>
>
> To achieve this, instead of focusing on diversifying kitchen maps, which often necessitates low-level control of agent motion, we have introduced several unique elements to enhance the complexity and diversity of the tasks:
>
> - **Enhanced Kitchen Layouts**: We have diversified the layout of kitchens by incorporating a wider range of **tools** (both in types and quantities) and **ingredients**. This approach differs from [1] and [2], where the emphasis is not on the variety of kitchen tools and ingredients. For example, in [1], there are only two types of ingredients and two types of tools. In [2], there are two types of ingredients, and two types of tools. In comparision, there are 27 unique ingredients, and 10 tools in CuisineWorld.
>
>
> - **Complex Task Design**: Our benchmark includes a broader spectrum of recipes, varying significantly in difficulty levels. This variation is not just in terms of the number of tools and ingredients required but also in the intermediate steps involved in each recipe. We invite you to refer to Figure 23 for a detailed illustration. This aspect of task complexity, particularly in the context of high-level planning, is not extensively explored in [1] and [2]. In [1,2] there are very limited number of dishes, 1 and 3 respectively. However, in CuisineWorld, there are 33 unique dishes.
>
>
> - **Multi-Dish Episodes and Collaborative Strategy Assessment**: We require agents to complete multiple dishes within a single episode, with the types of dishes varying to challenge and assess the collaborative strategies of the agents. Our level design ensures that there are shared intermediate steps among the types of dishes in a single episode. The system is tasked with multiple different goals at the same time. This approach allows us to use metrics like 'Collaborative Score' (CoS) to evaluate how agents collaborate to achieve higher dish throughput. This dynamic aspect of collaboration, especially in the context of dish expiration and shared tasks, offers a new dimension to the study of multi-agent cooperation, which is distinct from the environments in [1] and [2]. In [1] the goal is to finish as many dishes as possible in a limited amount of time. In [2], the goal is to finish one dish in the least amount of time. Both of them do not consider the density of the tasks (interval between dish orders coming to the kitchen) and its effect on coordination. As we mentioned earlier, this concpet (changing density of the tasks and measuring collaboration proficiency upon it) is at heart of CuisineWorld and our CoS metric, and it has demonstrated its effectiveness of benchmarking **collaboration between LLMs and human-NPCs** (as indicated in the abstract).
>
> > Table 12 made a good summary of related benchmarks though some related embodied multi-agent cooperation benchmarks with most of the features in the table are missing, such as [3][4].
>
> Thanks for pointing this out. We updated the paper and added the missing references.

---

> > ### Author Response · Authors · 2023-11-21
> > **Response to Reviewer YZDa 2/N**
> >
> > > While CuisineWorld boasts an appealing visual design, it does not directly align with the paper's experiments. The paper primarily relies on text-based states and interactions (Figure 26, even during human experiments, where humans are restricted to a textual interface with lots of text description of the game state)
> >
> > We appreciate your observation regarding the use of text-based states and interactions in our experiments, as contrasted with the visually appealing design of CuisineWorld. You are correct in noting that our experiments primarily utilize text-based interfaces for thinking and coordination processes, which are then projected into the virtual world of the CuisineWorld game. This approach is indeed in line with several seminal works in the field, such as Voyager[1] and DEPS[2], where Large Language Model based Autonomous Systems operate primarily through text-based interactions due to the current limitations in powerful vision-language models that can process visual data directly.
> >
> > However, it's important to clarify that this text-based approach does not restrict human players to a purely textual interface. In fact, as demonstrated in our Minecraft experiment, human players can engage with the game in a more immersive Virtual Reality (VR) interface, enjoying natural gameplay while interacting with agent counterparts. This aspect of our research showcases the versatility of our infrastructure in accommodating different modes of human-agent interaction, catering to both research needs and user experience. The video demo can be found in the submitted supplementary materials.
> >
> >
> > [1] Voyager: An Open-Ended Embodied Agent with Large Language Models
> >
> > [2] Describe, Explain, Plan and Select: Interactive Planning with Large Language Models Enables Open-World Multi-Task Agents
> >
> >
> > > In section D.2, the statement "In this game, the actions of human players can be obtained from an inverse dynamic model by checking pre-conditions and post-effects" is confusing to me. How was Figure 22 obtained? It seems to be in real-time with human players using the keyboard to move, how's the time step defined here and the "goto" action for the LLM agent implemented?
> >
> > Thanks for pointing this out. This due to the differnce between the two versions of CuisineWorld, text-based game and the real-time VR game that human can play in VR. Unlike the step-by-step nature of the text-based version of CuisineWorld, the real-time VR game operates continuously. To align the LLM’s processing with this dynamic environment, we implemented a system where the LLM checks user actions at regular intervals during the game loop, referred to as "time steps." These time steps are defined as 0.1 seconds. Then we can ensure LLM can respond to user actions in a timely manner, matching the pace of the real-time game.
> >
> > In the real-time version, human players control their agents directly through keyboard inputs, resulting in low-level actions like moving or picking up items. However, the LLM-agent operates on a higher, more temporally-extended level of atomic actions. To bridge this gap, when the LLM checks user actions, it doesn't just read the keyboard inputs. Instead, it assesses changes in the high-level game state, such as the agent’s location, and the status of tools and ingredients. This assessment allows the LLM to infer the temporally extended actions that align with its text-based decision-making process. Implementing this required a simple inverse dynamics model through checking state changes to translate low-level actions into high-level atomic actions, facilitating a seamless transition from the text game to the real-time virtual game.
> >
> > Regarding the specific query about the 'GoTo' action, we utilized the A* pathfinding algorithm, integrated into the Unreal Engine, to facilitate this movement.
> >
> > We hope these updates and clarifications provide a better understanding of how we adapted the CuisineWorld game to a real-time virtual environment, and how we managed the differences in user and LLM-agent interactions between the text-based and real-time VR versions.

---

> > > ### Author Response · Authors · 2023-11-21
> > > **Response to Reviewer YZDa 3/N**
> > >
> > > > The benchmark predominantly employs a "new auto-metric collaboration score CoS for assessing the collaboration efficiency". However, this metric is defined as the average task success rate with different time intervals for each dish that appears highly tailored to the specific environment and lacks a clear connection to "cooperation efficiency".
> > >
> > > We thank you for pointing this out. Below, we start with some intuition on why CoS makes sense in CuisineWorld, then we will clarify how can this generalize to other domains as well:
> > >
> > > Rationale of CoS:
> > >
> > > - In the kitchen scenario as demonstrated in CuisineWorld, hypothetically, when the dish order come very rarely (with a large interval), no matter if there is any collaboration, high success rate can easily attained as there is sufficient time.
> > >
> > > - However, as we reduce the interval, more and more dish order are flooding in. If the agents (and humans) are able to collaborate well, the productivity will be high, or namely, they can still manage to maintain a decent success rate. On the contrary, teams with poor collaborate will likely suffer from a substantial drop on success rate as the interval gets smaller. The same collapse will ultimately happen to a good collaborative team too when the interval gets too small, but good collaboration can always sustain longer.
> > >
> > > - Therefore, it makes sense to use the averaged success rate across different intervals as an indicator of the collaboration proficiency. More importantly, such metric asks for a game setting where the dish order will keep coming, in a changing interval, which also aligns with the original Overcooked! game experiences.
> > >
> > > How to generalize to other games:
> > >
> > > - We've demonstrated that CoS is able to generalize to Minecraft (see table 5). We simply modify the interval of the crafting tasks and therefore measure the collaboration efficiency there.
> > >
> > > - In more general multi-agent gaming scenarios, as long as the goal of the game itself includes completing some tasks (which we believe could be a generally applicable point), CoS can be applied -- the productivity of a good team should be able to handle both when there is very few tasks or many more and therefore, CoS could measure their collaboration proficiency.
> > >
> > > [1] On the Utility of Learning about Humans for Human-AI Coordination
> > > [2] Too many cooks: Bayesian inference for coordinating multi-agent collaboration
> > >
> > > > The absence of a train/test split is concerning, as prompt engineering can substantially impact performance. A thorough understanding of how the prompt is tuned for different tasks is crucial.
> > >
> > > We appreciate the reviewer's concern regarding the absence of a traditional train/test split and the potential impact of prompt engineering on performance. Our approach with the CuisineWorld dataset is indeed unconventional, and we align with recent research efforts in probing datasets, such as the study highlighted in [1].
> > > Our primary objective with CuisineWorld is to establish a grand challenge for LLM-based multi-agent collaboration and emergent gaming interaction. In this context, the entire dataset serves as a probing tool, intended to explore the capabilities and limitations of LLMs in complex, dynamic environments. We consciously chose not to enforce a train/test split to avoid limiting the scope of exploration and experimentation within the dataset. That being said, we do not encourage prompt engineering efforts that are specific for tweaking our benchmark but we believe openly inviting all avenues of approach on tackling this challenge can spark more interesting ideas and findings.
> > >
> > >
> > > [1] Winoground: Probing Vision and Language Models for Visio-Linguistic Compositionality

---

> ### Author Response · Authors · 2023-11-21
> **Response to Reviewer YZDa 4/N**
>
> > From C.3.4, level 3 has only two similar recipes which differ only with the words "salmon" and "tuna", only one demonstration may decrease the task difficulty significantly for the LMs, raising doubts about the benchmark formation.
>
> We appreciate your concern about the perceived task simplicity in Level 3 of CuisineWorld, where the recipes differ only in terms of "salmon" and "tuna." However, we respectfully disagree with the notion that a single demonstration significantly decreases task difficulty for LLMs. Our benchmark's design is not centered around creating complex recipes for LLMs to master, but rather on introducing and evaluating the complexity of collaboration through carefully designed recipe combinations.
>
> In Level 3, the challenge for LLMs is to effectively coordinate agents around shared tasks and bottleneck resources. A strategic collaboration example could involve assigning one agent to prepare cooked rice—a common component in all dishes—while others handle different kitchen stations, such as mixers or chopping boards. This requires the LLMs to identify and manage shared resources and task bottlenecks, and to establish an efficient “pipeline” for task execution, which significantly enhances throughput. Such strategic planning and coordination cannot be trivially inferred from a single demonstration.
>
> Furthermore, it is important to note that the demonstrations provided to the LLMs are based on Level 0 tasks, which are fundamentally different from those in Level 3. Therefore, the LLMs are essentially working on novel tasks that require adaptive and strategic thinking beyond replicating demonstrated actions.
>
> This aspects of our design underscores the benchmark's focus on assessing high-level planning and collaboration, rather than recipe complexity.
>
> > In section 5.1, The paper claims "more agents will lead to higher collaboration efficiencies. Thus, indicating that the LLM dispatcher can coordinate additional agents to execute tasks more efficiently". However, from Table 1, 4 agents perform worse than 3 agents in most scenarios, and even 2 agents achieve the highest score in levels 2 and 4, which contradicts the claim. It may provide more insights if the paper can provide more analysis on these contradictory results instead of only ablating on level_3, which seems to be the only level that "looks normal" (a.k.a 2 agents < 3 agents < 4 agents with somewhat clear gaps)
>
> Thank you for highlighting the discrepancies in our results concerning the relationship between the number of agents and collaboration efficiency. We have revised our claim to more accurately reflect that while generally increasing the number of agents can lead to higher efficiency, there are scenarios where adding more agents may actually hinder performance.
>
> Our revised claim now acknowledges that the relationship between the number of agents and task efficiency is not always linear and can be influenced by various factors, including task complexity and the nature of the agents' collaboration. We have added detailed discussions around this revised claim to explore these nuances and provide a more comprehensive understanding of when and why more agents may or may not lead to better performance.
>
> In our initial study, we chose Level 3 for our ablation study because it showed a clear pattern where increasing the number of agents improved performance. We've updated our paper to explain that this choice was based on the observation that the LLM performed well on this level, offering a stable platform to assess the impact of varying agent numbers. When LLMs do not perform well on certain levels, additional unknown factors might confound the results, making it challenging to isolate the effects of ablated factors on performance.
>
> In response to your suggestion, we have conducted a preliminary qualitative analysis and found the following failure mode:
>
> 1. Inability to prioritize the task order. Sometimes LLM will skip the task that is on the top of the task queue and leads to task expiration. Therefore, the success rate will drop.
>
> 2. Unable to understand occupy() state instruction. In our CuisineWorld, agents need to wait for a varying timesteps before the cooking is done. The wait time depends on the specific tool. When agents attempt to take out ingredients immediately after activating the tool, actions will fail. Instead of keeping waiting, the agents might work on other task components which will slow down the overall progress.
>
> 3. Unable to allocate agents to the correct subgoals when there are multiple concurrent dishes. This is especially the case when there are 4 agents.

---

> > ### Author Response · Authors · 2023-11-21
> > **Response to Reviewer YZDa 5/N**
> >
> > > Table 4, "For a fair comparison, all tests employed identical prompt inputs" using "identical" prompt for different LM families may not be "fair", especially if the prompt is "tuned" specifically for one model. More details on the prompt engineering process may help clarify these concerns.
> >
> > Thank you for your comment regarding the use of identical prompts across different Language Model (LM) families in Table 4. We understand your concern about the fairness of this approach, particularly if a prompt is more optimized for one model over others. In response, we have updated our paper with more detailed information about our prompt engineering process.
> > Our initial approach began with a simple and generic prompt, designed to be as neutral as possible. This baseline prompt was tested across all models, including GPT-4, Claude, and other LLMs in our study.
> >
> > We observed that only GPT-4 and Claude were able to generate reasonable planning responses with this initial prompt. Consequently, we refined our prompt engineering efforts to better suit these two models, aiming to optimize their performance and response quality.
> >
> >
> > For the other LLMs, we also attempted individual prompt engineering. However, these efforts did not yield significant improvements in their responses compared to the initial trial. After several rounds of testing and refinement, we concluded that further prompt customization for these models did not result in notably better outcomes.
> > As a result, we decided to use a consistent prompt across all LLMs for the final comparison. This decision was made to maintain uniformity in testing conditions, despite the prompts being more closely tuned to GPT-4 and Claude. We believe this approach offers a reasonable balance between fairness and practicality in evaluating the capabilities of different LLMs under similar conditions.
> >
> >
> > We hope this explanation clarifies our methodology and addresses your concerns regarding prompt engineering and fairness in our comparison. We are open to further discussion on this matter and welcome any additional thoughts or suggestions you may have.
> >
> > > Table 3 presents a perplexing scenario where four agents using a two-agent demo outperform four agents using a four-agent demo, albeit marginally. This result could benefit from a clearer explanation.
> >
> > Thank you for pointing out the intriguing result in Table 3 where four agents using a two-agent demonstration outperform those using a four-agent demonstration, albeit by a marginal degree. We have revised our analysis of these results to provide a clearer understanding.
> >
> > The primary focus of the results presented in Table 3 is that GPT-4 can leverage demonstration of a different number of agents to perform task planning and allocations for a team size, which remains valid and is not contradicted by the observed anomaly.
> >
> > However, we understand your concern that the comparison between GPT-4 leverags two-agent and four-agent demo could be a bit counterintuitive. Therefore, we visualize the result, here is what we found:
> >
> > -  This might due to the fact that the task structure is friendly to a two agent or four agent team, which means with four agents, LLM can divide the agnets into two teams of two agents, with each team operating independently. This observation also highlight the model’s capability to perform task allocations and task plannings for more agents than if has previously seen.

---

> > > ### Author Response · Authors · 2023-11-21
> > > **Response to Reviewer YZDa 6/N**
> > >
> > > > Novel game adaptation of Minecraft seems very promising, but the details are extremely limited. It would be more convincing if there were more details and formal experiments on it. For example, how is the "adaptation" conducted? What's the additional human effort required (such as prompt engineering and rounds of re-playing for prompt tuning)?
> > >
> > > We appreciate your interest in our novel adaptation of Minecraft and acknowledge the need for more detailed information. Our approach to adapting Minecraft for our experiments was indeed multifaceted, encompassing both adjustments to the Language Model (LLM) and developments on the game development side.
> > >
> > > 1. **LLM Modifications**: On the LLM side, we focused on updating the background knowledge of the model. This included:
> > >
> > >     - **Action Space Explanation**: We provided the LLM with detailed information about the possible actions and interactions within the Minecraft environment. This step was crucial to align the model's understanding with the game's mechanics.
> > >
> > >     - **Recipe and Tool Definitions**: We also included comprehensive definitions of recipes, tools, and ingredients specific to Minecraft. This enriched the LLM's contextual knowledge, enabling it to make more informed decisions within the game.
> > >
> > >     We believe these modifications are reasonable and necessary, as without these knowledge, it’s very difficult for model to operate in an unknown environments.
> > >
> > >
> > > 2. **Game Development Adjustments**: On the game development side, we dedicated efforts to:
> > >
> > >     - **Text-to-Game Interaction Translation**: We developed code to translate text-based interactions and commands from the LLM into actionable inputs within the virtual game environment. This translation layer was key to bridging the gap between the LLM's text-based outputs and the game's interactive elements.
> > >
> > >
> > > We intentionally avoided extensive prompt engineering for adaptation to the new game domain. Except for the modified background knowledge mentioned above, our approach did not involve iterative prompt tuning processes. We believed that keeping prompt engineering to a minimum would provide a more authentic and generalizable understanding of the LLM's capabilities in adapting to new environments.
> > >
> > > We have updated the paper to include more details and we thanks the reviewer for pointing this out.

---

> > > > ### Author Response · Authors · 2023-11-21
> > > > **Response to Reviewer YZDa 7/N**
> > > >
> > > > > The method employed in the paper primarily relies on intensive prompt engineering, without introducing novel designs for multi-agent cooperation. It's well known that LLMs can perform "in-context learning", providing demonstrations and reasoning steps can help improve performance, there's nothing new to take away from the method and experiments. As mentioned above, the "emergent ability" of MindAgent is not well supported by the results. Providing only screenshots of part of the prompts as in Figures 6 and 7 is not enough. More details on the full prompt and game episode may help clarify these concerns.
> > > >
> > > > Thank you raising this. Here, we will first try to address your concern on "in-context learning" and take-away messages, then move to the emergent ability and prompting details.
> > > >
> > > > In-context learning and take-away message:
> > > >
> > > > - As we've illustrated in Fig. 6 and 7 in the supplementary, the demonstration we provide to the LLM **is always on the same simple task** across all level of games -- no matter how complex the dishes and collaboration regime within the current level can be, the demonstration, is always on a simple task with one dish "porkMeatCake", which is very similar to "salmonMeatCake", a dish that is categorized as "level_0" or "very simple" in Figure 8.
> > > >
> > > > - Compared to the previous in-context learning setting in LLMs, VLMs, where they simply learn from examples of the same task with similar complexity, the ability of **generalizing** from a demonstration on such simple task to more complex dishes/levels, with very simple prompt including game rules and recipies only, is rarely explored, especially in LLM planning and multi-agent collaboration. Our experiments conduct a comprehensive exploration on just that. We believe this can be one take-home message from our result.
> > > >
> > > > - Obviously, there are many more take-home messages as well, including how many agents the current LLMs are able to robustly coordinate under what complexity of tasks (from our main results), to which extent can LLMs generalize from a demonstration of fewer agents (from results in table 3), how does our proposed prompting techniques affect the collaboration (from ablations in table 2), how does different LLMs perform (from results in table 4), etc.
> > > >
> > > > Prompting details:
> > > >
> > > > - Figure 7 \& 8 show the **complete** prompt we use, there is indeed nothing more -- just game rules, recipes of the current level, and one-shot demonstration from a "very simple" portMeatCake task (yes, we use the same task as in-context demonstration for all levels). This is the **full prompt**. We're sorry for not making this clear. In any case, please let us know if there are further details on the prompt we need to clarify.
> > > >
> > > > Emergent ability:
> > > >
> > > > - We thank your for your suggestion. To visulize the emergent collaboration from our designed prompt (only rules, recipes, and one-shot demonstrations on a much simpler task), we've listed some gaming interaction in appendix H for cooking burgers. Considering the length of the full game log, we only show the interaction until the first dish is fulfilled. The full log can be found [here](https://drive.google.com/file/d/1pLViwnC_ngL4cArDnjhLqVchZKZ-6sp6/view?usp=sharing).

---

> > > > > ### Comment · Reviewer_YZDa · 2023-11-23
> > > > >
> > > > > Thanks for the detailed responses! I highly appreciate it and some of my concerns are addressed. However, my main concern remains: the claims made in this paper are not well supported by the experiments and results provided, and there are not many new efforts in the revision to improve this. Therefore I'm keeping my score for this version of the paper.

---

> ### Author Response · Authors · 2023-11-23
> **Response to Reviewer YZDa, 8/N**
>
> >  "Thanks for the detailed responses! I highly appreciate it and some of my concerns are addressed. However, my main concern remains: the claims made in this paper are not well supported by the experiments and results provided, there are not many new efforts in the revision to improve this."
>
> We thank you for the constructive comments and helpful suggestions again. We address your main concerns below and describe every change that was made to the manuscript in our revision.
>
> > The CuisineWorld, based on the video game Overcooked which is popular for measuring multi-agent cooperation and has many existing environments already[1][2]. The benchmark introduces more cuisines but falls short in diversifying kitchen maps when compared to previous Overcooked environments.
>
> We addressed this comment in red-ink of Appendix Sec. E.1 in the new revision paper.
>
> > In section D.2, the statement "In this game, the actions of human players can be obtained from an inverse dynamic model by checking pre-conditions and post-effects" is confusing to me. How was Figure 22 obtained? It seems to be in real-time with human players using the keyboard to move, how's the time step defined here and the "goto" action for the LLM agent implemented?
>
> We addressed your comments in red-ink of Appendix Sec. E.2. (we add one more section in the appendix of the revision, so the Sec. E.2 maps to the previous Sec. D.2 as per your comments.)
>
> > The benchmark predominantly employs a "new auto-metric collaboration score CoS for assessing the collaboration efficiency". However, this metric is defined as the average task success rate with different time intervals for each dish that appears highly tailored to the specific environment and lacks a clear connection to "cooperation efficiency".
>
> We addressed these comments in red-ink of Appendix Sec. E.6 in our revision.
>
> > In section 5.1, The paper claims "more agents will lead to higher collaboration efficiencies. Thus, indicating that the LLM dispatcher can coordinate additional agents to execute tasks more efficiently". However, from Table 1, 4 agents perform worse than 3 agents in most scenarios, and even 2 agents achieve the highest score in levels 2 and 4, which contradicts the claim. It may provide more insights if the paper can provide more analysis on these contradictory results instead of only ablating on level_3, which seems to be the only level that "looks normal" (a.k.a 2 agents < 3 agents < 4 agents with somewhat clear gaps)
>
> We addressed your comments in red-ink of Section 5.1 findings.
>
> > Table 4, "For a fair comparison, all tests employed identical prompt inputs" using "identical" prompt for different LM families may not be "fair", especially if the prompt is "tuned" specifically for one model. More details on the prompt engineering process may help clarify these concerns.
>
> We describe the detailed prompt engineering process in red-ink Section B of our revision.
>
> > Table 3 presents a perplexing scenario where four agents using a two-agent demo outperform four agents using a four-agent demo, albeit marginally. This result could benefit from a clearer explanation.
>
> We addressed your comments in red-ink of Section 6.2 of our revision.
>
> > Novel game adaptation of Minecraft seems very promising, but the details are extremely limited. It would be more convincing if there were more details and formal experiments on it. For example, how is the "adaptation" conducted? What's the additional human effort required (such as prompt engineering and rounds of re-playing for prompt tuning)?
>
> We addressed your comments in red-ink of Appendix Sec. F.1 for our revision.
>
> > The method employed in the paper primarily relies on intensive prompt engineering, without introducing novel designs for multi-agent cooperation. .... ....... Providing only screenshots of part of the prompts as in Figures 6 and 7 is not enough. More details on the full prompt and game episode may help clarify these concerns.
>
> We addressed this issue in Appendix Section H and Section B. We add parts of the log as "The full interaction history for finishing one dish in level 9", which is shown in Sec. H. and we provide the full game log  [here](https://drive.google.com/file/d/1pLViwnC_ngL4cArDnjhLqVchZKZ-6sp6/view?usp=sharing) for your reference. Due to the page limitaion, we can't attach the full game log in our revised paper. Hope for your understanding.
>
> > Table 12 made a good summary of related benchmarks though some related embodied multi-agent cooperation benchmarks with most of the features in the table are missing, such as [3][4].
>
> We compare our work vs [3][4] in the Table 13 of Appendix Section B1 with red-ink, and we cited the papers and discussed the differences with our work in line 3, line4, line 5, line 11, and line 12 in the section 2 as well as our revised manuscript .

---

> ### Author Response · Authors · 2023-11-23
> **Response to Reviewer YZDa, 9/N**
>
> > Thanks for the detailed responses! I highly appreciate it and some of my concerns are addressed. However, my main concern remains: the claims made in this paper are not well supported by the experiments and results provided, and there are not many new efforts in the revision to improve this. Therefore I'm keeping my score for this version of the paper.
>
> Thanks for the thoughtful comments. Furthermore, we also conducted additional reinforcement learning experiments. Results and comparison with our MindAgent results are be added in the Appendix Table 12 for supporting our claims.
>
> We really appreciate your very helpful suggestion and all your constructive feedback. We addressed each of your comments in our revised manuscript for your reference.

---

### Official Review · Reviewer_RwfK · 2023-10-31

**Soundness:** 4 excellent
**Presentation:** 3 good
**Contribution:** 3 good
**Rating:** 8
**Confidence:** 3

**Summary:**

This paper proposes an infrastructure, MindAgent, for evaluating planning and coordination capabilities in the context of gaming interaction. To facilitate multi-agent planning capabilities of LLMs, the paper designs an effective set of prompt templates, memory modules, and state and action processing modules. Additionally, the paper reformulates the optimization objectives and constraints of multi-agent planning into natural language descriptions and uses them as prompts to guide LLM planning. The paper also introduces a virtual kitchen game called CuisineWorld as a benchmark for LLM-based multi-agent planning. Furthermore, the paper evaluates the planning capabilities of various LLMs in multi-agent collaboration tasks and human-agent collaboration tasks in CuisineWorld using MindAgent.

**Strengths:**

***Originality & Significance***

Although this work is mainly application-oriented for LLMs, its novelty lies in proposing an infrastructure to explore the potential of LLMs in multi-agent planning and conducting experiments on multiple LLMs. I believe this work will inspire the LLM community and provide a valuable test bed for evaluating LLM capabilities.

***Quality & Clarity***

This work provides detailed descriptions and examples of the components of MindAgent. The paper also offers a good description of the environment setup and level settings in CuisineWorld. Furthermore, the paper validates the abilities of multiple LLMs, such as GPT-4, through multi-agent collaboration and human-agent collaboration tasks, and provides detailed experimental settings. I think the paper is clear and of high quality.

**Weaknesses:**

Please refer to the questions section.

**Questions:**

1. Have the authors considered incorporating a human-agent communication module in MindAgent? As suggested by Gao et al. [1], introducing interpretable human-agent communication into collaborative games can effectively improve human-agent collaboration performance and human subjective preferences. Natural language is the best medium for human-agent communication, which is also a natural advantage of LLM-based agents in human-agent collaboration.

2. Can the design of MindAgent support collaborative games that require more domain-specific knowledge? For example, Multiplayer Online Battle Arena (MOBA) games [1], First-person Shooter (FPS) games [2], and Diplomacy [3] have very complex gameplay and their outcomes heavily depend on the planning and collaboration capabilities of the agents. The authors could discuss how the infrastructure needs to be modified when extending it to other games, which would enhance the generalisability of the infrastructure.

---

[1] Gao, Yiming, et al. Towards Effective and Interpretable Human-Agent Collaboration in MOBA Games: A Communication Perspective. ICLR. 2023.

[2] Jaderberg, Max, et al. Human-level performance in 3D multiplayer games with population-based reinforcement learning. Science. 2019.

[3] FAIR, et al. Human-level play in the game of Diplomacy by combining language models with strategic reasoning. Science. 2022.

---

> ### Author Response · Authors · 2023-11-21
> **Response to Reviewer RwfK**
>
> > Have the authors considered incorporating a human-agent communication module in MindAgent? As suggested by Gao et al. [1], introducing interpretable human-agent communication into collaborative games can effectively improve human-agent collaboration performance and human subjective preferences. Natural language is the best medium for human-agent communication, which is also a natural advantage of LLM-based agents in human-agent collaboration.
>
> Thank you for highlighting the potential of incorporating human-agent communication in MindAgent, as suggested by Gao et al. [1]. We agree that introducing interpretable human-agent communication in collaborative games is a promising direction for enhancing human-agent collaboration. Accordingly, we have updated our paper to include citations and discussions related to [1], acknowledging its relevance to our work.
>
>
> We adopted a straightforward method for facilitating communication between the human user and the LLM-based agent. Participants in the experiment were instructed to verbally express their intents or comments. These verbal inputs were then transcribed and integrated into the LLM’s context as “current human instruction: <instruction>”.
>
>
> Implementation and Results: This simple yet effective approach allowed for real-time adjustments in the behavior of the LLM-based agents based on human inputs. For instance, in our demo videos can be found in the supplementary videos, when a user suggested a specific strategy or highlighted a priority task, the agent was able to promptly adapt its actions accordingly. This led to a more dynamic and responsive collaboration between humans and agents, showcasing the potential of natural language communication in enhancing the collaborative experience.
>
>
> In summary, while our initial trials in human-agent communication are relatively basic, they clearly demonstrate the effectiveness of integrating natural language communication in LLM-based collaborative environments. We plan to further develop and refine this aspect of our system in future work, drawing inspiration from the findings of Gao et al. [1] and other seminal works in the field.
>
>
> > Can the design of MindAgent support collaborative games that require more domain-specific knowledge? For example, Multiplayer Online Battle Arena (MOBA) games [1], First-person Shooter (FPS) games [2], and Diplomacy [3] have very complex gameplay and their outcomes heavily depend on the planning and collaboration capabilities of the agents. The authors could discuss how the infrastructure needs to be modified when extending it to other games, which would enhance the generalisability of the infrastructure.
>
>
> We thank the reviewer for suggesting the exploration of domain-specific knowledge in collaborative games like MOBA, FPS, and Diplomacy, and its implications for the design of MindAgent. Our current work in CuisineWorld has indeed laid the groundwork for such adaptations, primarily through the use of In-Context Learning (ICL).
>
> In CuisineWorld and Minecraft, we inject domain-specific knowledge (such as recipes) directly into the context, which the Large Language Model System utilizes to inform its decision-making and collaboration strategies. This demonstrates the feasibility of adapting MindAgent to other domains where specific knowledge plays a crucial role.  In addition, techniques like Retrieval-Augmented Generation (RAG) and Fine Tuning could be pivotal in further developing MindAgent's capabilities to handle such domain-specific complexities.
>
> Thanks for the suggestions, and we have updated the paper accordingly.

---

### Official Review · Reviewer_DUbr · 2023-11-01

**Soundness:** 2 fair
**Presentation:** 3 good
**Contribution:** 2 fair
**Rating:** 3
**Confidence:** 4

**Summary:**

This work proposes an infrastructure for LLM-based agents to perform task distribution across a number of agents. It focus on planning and coordination ability of LLM with in-context learning from a few examples. The work tests on a multi-agent gaming environment, CuisineWorld, as well as a few other multi-agent collaboration environments, and obtain promising results.

**Strengths:**

Contributions:

1. Evaluate LLM agent in a multi-agent gaming environment; test LLM's ability to serve as a task allocator.

2. Proposes a collaboration score to evaluate and benchmark coordination agents.

**Weaknesses:**

1. Technical novelty: The technical novelty of this work is lacking because the multi-agent setting is really just adding an LLM call to allocate what different agents should do. Concretely, it is a simple prompting and what is supposed to be an optimization problem is all packed into one LLM call. This doesn't seem to be a very principled way of studying the agent allocation.

2. Problem setting: the multi-agent collaboration problem is reduced to a top-down allocation: a distributor distributes tasks for agents to do. But for a collaboration problem to work, there should also be communication between the agents, which this work does not study.
- the title is also quite misleading: it is titled "emergent interaction" however, there is no real interaction between the agents, i.e., communication of agent's individual's ability, limitation, progress etc. It is more of a allocation / coordination problem of a central agent.

3. Experimental results: The work evaluate on Minecraft environment, but there has been quite a few LLM-based Minecraft agents, and the work does not offer comparison between this method and existing works as baselines.

**Questions:**

1. What are the common failure cases of the LLM agent allocation? Under what environment / optimization constraints would the agent fail to allocate works correctly?

---

> ### Author Response · Authors · 2023-11-21
> **Response to Reviewer DUbr 1/N**
>
> > Technical novelty: The technical novelty of this work is lacking because the multi-agent setting is really just adding an LLM call to allocate what different agents should do. Concretely, it is a simple prompting and what is supposed to be an optimization problem is all packed into one LLM call. This doesn't seem to be a very principled way of studying the agent allocation.
>
> Thank you for the comments. First of all, we should point out that **the motivation of this paper is not about proposing a set of agent allocation tasks to compare planning proficiency between LLMs and other canonical planning methods (ex. what [1] did)**, but the following (**in our abstract**):
>
> ```
> However, despite the introduction of numerous gaming frameworks, the community lacks adequate benchmarks that support the implementation of a general multi-agent infrastructure encompassing **collaboration between LLMs and human-NPCs**. We propose a novel infrastructure-MindAgent—for evaluating planning and coordination capabilities in the context of gaming interaction.
> ```
>
> That is, we simply would like to explore whether LLMs, which enjoys a powerful and native natural language interface that can interact with humans, can support sohphsticated multi-agent collaboration and human-NPC collaboration tasks, and this results in our benchmark. We've made this very clear -- the subject of study is **collaboration between LLMs and human-NPCs**.
>
> That being said, we do agree that showing some results of the canonical planning approach, including RL, could offer a better understanding of the dynamics of the collaboration tasks. Below, we provide some results of using a canonical RL algorithm on the proposed benchmarks for LLMs:
>
> We implemented a masked version of Proximal Policy Optimization (PPO) approach. In additon, we provide heavily engineered dense reward. Despite leveraging privileged information (masked action, what action is valid during the execution), the RL strategies show limited success, even in two-agent settings. For level 1, PPO achieve 45.14% success rate and for level 4, PPO achieves 59.8% for \tau=2.5 after training for 30M steps or 500, 000 episodes. In comparison, GPT-4 agent achieves 95%  success rate on both settings.
>
> Further the trained RL policy cannot adapt to varying numbers of agents and to new tasks, or even a change in the number of ingredients. This limitation underscores the complexity and dynamic nature of our environment.
>
> [1] https://arxiv.org/abs/2302.06706
>
> > Problem setting: the multi-agent collaboration problem is reduced to a top-down allocation: a distributor distributes tasks for agents to do. But for a collaboration problem to work, there should also be communication between the agents, which this work does not study.
>
> We thank the reviewer for pointing out that our method is a top-down allocation. The multi-agent task planning problem [1], is a very difficult problem when they are multiple dependencies.  We want to address that is different from previous settings as in [1], where the task is known priori. In our settings, the tasks are not known prior to the agents simulating the experience of a real kitchen, dish orders keep flooding in. This modification will only push the original problem harder.
>
> We acknowledge that communication between agents is indeed a critical aspect of collaboration, and our current model does not explicitly study inter-agent communication. This is a limitation of our study, and we believe it represents an exciting avenue for future research. The integration of direct agent-to-agent communication would undoubtedly add a rich layer of complexity and realism to the collaboration model, offering insights into emergent behaviors, negotiation strategies, and decentralized decision-making processes.
>
> However, this addition will also add communication cost which might not be ideal for gaming applications where a faster response is desired. Therefore, in this specific work (MindAgent) that focuses on **collaboration between LLMs and human-NPCs** in **gaming scenarios**, we choose to anchor our study on a centralized planning scheme.
>
> [1] Korsah et al. A comprehensive taxonomy for multi-robot task allocation.

---

> > ### Author Response · Authors · 2023-11-21
> > **Response to Reviewer DUbr 2/N**
> >
> > >  the title is also quite misleading: it is titled "emergent interaction" however, there is no real interaction between the agents, i.e., communication of agent's individual's ability, limitation, progress etc. It is more of a allocation / coordination problem of a central agent.
> >
> > We appreciate the reviewer's feedback regarding the title of our paper and understand the concerns raised about the term "emergent interaction." To clarify, our use of "interaction" in this context is twofold:
> >
> > - **Interaction with the Environment**: We emphasize the ability of agents to interact dynamically with the environment. This interaction does not require specialized training for specific tasks, showcasing emergent behaviors.
> >
> > - **Interaction with Human Gamers**: Another crucial aspect of interaction we explore is the collaboration and teaming with human players. Our agents are designed to adapt to human strategies and work alongside human gamers efficiently, achieving commendable performance without the need for specialized training to understand human gameplay.
> >
> > > Experimental results: The work evaluate on Minecraft environment, but there has been quite a few LLM-based Minecraft agents, and the work does not offer comparison between this method and existing works as baselines.
> >
> > We appreciate the reviewer’s insight regarding the comparison with existing LLM-based Minecraft agents. It is important to note that our work operates within a distinctly different framework compared to the referenced LLM-based agents in Minecraft. Our focus is on Multi-Agent Systems (MAS), which inherently introduces different challenges and dynamics compared to single-agent environments. This distinction in the setting - MAS versus single-agent systems - is crucial as it changes the nature of the problem and the applicable solutions.
> >
> > Despite the different settings, we have indeed incorporated some techniques from existing single-agent LLM-based works into our approach, for example leveraging feedback, incorporating demonstrations etc, which has been adopted by some seminal and concurrent work on LLM + gaming [1,2].
> >
> > [1] Describe, Explain, Plan and Select: Interactive Planning with Large Language Models Enables Open-World Multi-Task Agents
> >
> > [2] JARVIS-1: Open-world Multi-task Agents with Memory-Augmented Multimodal Language Models

---

### Official Review · Reviewer_tSpT · 2023-11-03

**Soundness:** 2 fair
**Presentation:** 2 fair
**Contribution:** 2 fair
**Rating:** 3
**Confidence:** 4

**Summary:**

The paper proposes a new benchmark CuisineWorld for evaluating the planning and coordination capabilities of LLMs in multi-agent settings. Different from previous research, this benchmark is characterized by (1) its incorporation of multi-task objectives, (2) the involvement of more than two agents, and (3) the utilization of a centralized system for coordination. In this context, an LLM functions as the centralized system. At every timestep, the LLM processes each agent's state along with a prompt including recipes, demonstrations, the fundamental rules of the game, and memory history, and outputs the optimal actions for the agents. The paper demonstrates that GPT-4 has a strong planning capabilities on the proposed benchmark.

**Strengths:**

- The creation and development of a new benchmark, entailing a substantial amount of effort, is a significant strength of the paper.
- The introduction of a novel metric, CoS (Coordination Score), to assess coordination capabilities is a noteworthy contribution.

**Weaknesses:**

- The proposed LLM coordination system, MineAgent, lacks novelty as its approach of leveraging scratchpad or memory has been extensively explored in prior research.
- The proposed environment features limited state and action spaces and furnishes all the requisite recipes to solve tasks, possibly oversimplifying the challenge. In this situation, heuristic or RL planners could be readily employed. However, the paper does not provide comparisons with these approaches.
- The proposed environment bypasses low-level control, further oversimplifying the problem. From my understanding, an agent can move to any location with a single action, without the need to consider spatial information. This eliminates the need for spatial reasoning in the LLM planner.
- The paper claims that the LLM can seamlessly adapt to new planning problems across different domains, but this assertion is questionable considering the dependence on context length. While the simplicity of the current setting allows all environment information to be described within 1K tokens, this approach may prove inadequate for more challenging environments. Additionally, engaging in manual prompt engineering to enhance performance may entail significant monetary costs.
- The description of the environment setting requires further clarification. Does $\tau_\mathrm{int, (1)}$ mean that a new task will be added at every timestep? What is the maximum horizon of an episode?
- The paper only highlights the emergent behavior of GPT-4 while neglecting to discuss its potential drawbacks. A more balanced perspective could be achieved by including examples of GPT-4's failure cases.

**Questions:**

See the weaknesses above.

---

> ### Author Response · Authors · 2023-11-21
> **Response to Reviewer tSpT 1/N**
>
> We sincerely thank you for your time and constructive comments. Below, we provide detailed replies to your comments and hope we can resolve your major concerns.
>
> > The proposed LLM coordination system, MineAgent, lacks novelty as its approach of leveraging scratchpad or memory has been extensively explored in prior research.
>
> First, we want to emphasize that compare to the generative agents, we are using a centralized coordination planning. In our work, we have a  complex combinatorial  action space,  each agent has a action space of 221 due to the presence of multiple ingredients, multiple tools available in the environment., and with four agents, the action space becomes 2385443281 due to exponential effect of adding more agents. In addition, our work explored sophisticated spatial-temporal reasoning. To efficiently collaborate, agents need to work on multiple different tasks at the same time, manage to avoid space conflict (two or more agents cannot work on the same tools), etc.
>
> In  addition, compared with concurrent work focus on planning (ex. [1]), we have more agents, more environments, and our environments are extensible and adaptable.
>
> Thirdly, we created a new benchmark with extensive experiment. We even perform cross-domain experiments to adapt our settings to Minecraft.
>
> [1] JARVIS-1: Open-world Multi-task Agents with Memory-Augmented Multimodal Language Models
>
>
> > The proposed environment features limited state and action spaces and furnishes all the requisite recipes to solve tasks, possibly oversimplifying the challenge. In this situation, heuristic or RL planners could be readily employed. However, the paper does not provide comparisons with these approaches.
>
> Thank you for the comments. First of all, we should point out that **the motivation of this paper is not about proposing a set of planning tasks to compare planning proficiency between LLMs and other canonical planning methods (ex. what [1] did)**, but the following (**in our abstract**):
>
> ```
> However, despite the introduction of numerous gaming frameworks, the community lacks adequate benchmarks that support the implementation of a general multi-agent infrastructure encompassing collaboration between LLMs and human-NPCs. We propose a novel infrastructure-MindAgent—for evaluating planning and coordination capabilities in the context of gaming interaction.
> ```
>
> That is, we simply would like to explore whether LLMs, which enjoys a powerful and native natural language interface that can interact with humans, can support sohphsticated multi-agent collaboration and human-NPC collaboration tasks, and this results in our benchmark. We've made this very clear -- the subject of study is **collaboration between LLMs and human-NPCs**.
>
> That being said, we do agree that showing some results of the canonical planning approach, including RL, could offer a better understanding of the dynamics of the collaboration tasks. Below, we try to address your concern on action space, alternative methods, etc:
>
> We respectfully disagree the reviewer's observation regarding the "limited state and action spaces" of our proposed environment. However, we argue that the complexity of our environment is not trivial due to the presence of multiple and duplicated tools and ingredients. In a single-agent scenario, the action space expands to 221, and with four distinct agents, it grows exponentially to approximately 2.39 billion. This immense action space presents a unique challenge specific to multi-agent research.
>
> Addressing the enormous action space, we implemented a masked version of Proximal Policy Optimization (PPO) approach. In additon, we provide heavily engineered dense reward. Despite leveraging privileged information (masked action, what action is valid during the execution), the RL strategies show limited success, even in two-agent settings. For level 1, PPO achieve 45.14% success rate and for level 4, PPO achieves 59.8% for \tau=2.5 after training for 30M steps or 500, 000 episodes. In comparison, GPT-4 agent achieves 95%  success rate on both settings.
>
> Further the trained RL policy cannot adapt to varying numbers of agents and to new tasks, or even a change in the number of ingredients. This limitation underscores the complexity and dynamic nature of our environment.
>
> [1] https://arxiv.org/abs/2302.06706

---

> > ### Author Response · Authors · 2023-11-21
> > **Response to Reviewer tSpT 2/N**
> >
> > > The proposed environment bypasses low-level control, further oversimplifying the problem. From my understanding, an agent can move to any location with a single action, without the need to consider spatial information. This eliminates the need for spatial reasoning in the LLM planner.
> >
> > Thank you for your insightful feedback. We acknowledge the reviewer's concern regarding the bypassing of low-level control in our environment, which indeed simplifies the problem to an extent. However, our research is intentionally focused on multi-agent high-level planning in gaming scenarios, a domain that presents substantial challenges in its own right [1], also as evidenced by our results where even the best agent achieved only a limited CoS score below 0.7.
> >
> > The complexity in high-level planning is underscored by the need for sophisticated spatial-temporal reasoning, such as resolving conflicts when multiple agents vie for the same location, task planning when there are multiple dependencies. Our work aims to push the boundaries of what Large Language Model-based  Multi-agent Systems (MAS) can achieve in this high-level coordination and planning context. By concentrating on this aspect, we contribute valuable insights into the capabilities and limitations of LLMs in complex, multi-agent environments.
> >
> > We have indeed explored the integration of low-level actions in our Minecraft experiment, underscoring our recognition of its importance. However, for the scope of the current work, especially in the Cuisineworld scenario, we consciously chose to focus on high-level planning. This decision aligns with our objective to deeply understand and advance LLM capabilities in coordinating multiple agents before introducing the additional complexities of low-level control.
> >
> > Furthermore, it's pertinent to note that our approach aligns with concurrent research in the field. For instance, in single-agent planning systems like DEPS [2] and JARVIS-1[3], the low-level control is managed through Reinforcement Learning (RL) or Imitation Learning (IL) controllers, while LLMs oversee high-level planning. This also appies to Voyager[4], where the low-level control is scripted with the help of MineFlyer, a library for automating agent actions. This division of labor, separating high-level planning from low-level control, has proven effective and is a recognized methodology in current research community.
> >
> > [1] Korsah et al. A comprehensive taxonomy for multi-robot task allocation.
> >
> > [2] Describe, Explain, Plan and Select: Interactive Planning with Large Language Models Enables Open-World Multi-Task Agents
> >
> > [3] JARVIS-1: Open-world Multi-task Agents with Memory-Augmented Multimodal Language Models
> >
> > [4] Voyager: An Open-Ended Embodied Agent with Large Language Models

---

> ### Author Response · Authors · 2023-11-21
> **Response to Reviewer tSpT 3/N**
>
> >  The paper claims that the LLM can seamlessly adapt to new planning problems across different domains, but this assertion is questionable considering the dependence on context length. While the simplicity of the current setting allows all environment information to be described within 1K tokens, this approach may prove inadequate for more challenging environments. Additionally, engaging in manual prompt engineering to enhance performance may entail significant monetary costs.
>
> We thank the reviewer for pointing out that context length might be a issue for general planning. Indeed, there could be more sophisticated tasks with more complicated and long interaction. But:
>
> 1. First of all, MindAgent aims at **benchmarking the multi-agent gaming interaction planning of LLMs and between LLMs and humans**. Per our results suggested, even for this relatively simplified scenarios with shorter interaction history and limited context-length requirement, the collaboration score can still low, meaning that the context-length is not the major concern given the current collaboration proficiency of LLMs.
>
> 2. Moreover, even from the perspective of how can LLMs handle the context length, the context length is becoming less of a issue as the current version of GPT-4 can handle 128k context tokens in a single prompt,the approximately equivalent of 300 pages of text as suggested by OpenAI[6], which could allow sufficient interaction history. Moreover, we don’t need to look at the full interaction history as is the case in our work. History trimming is a common practice  as in [4,5] and can further mitigate the context length limit. In addition, for tasks with interaction that could be extremely complicated, we can optional use memory-based approach as in [1,2,3].
>
> [1] Retrieval augmented generation for knowledge-intensive nlp tasks
>
> [2] Generation-augmented retrieval for open-domain question answering
>
> [3] JARVIS-1: Open-world Multi-task Agents with Memory-Augmented Multimodal Language Models
>
> [4] Compressing Context to Enhance Inference Efficiency of Large Language Models
>
> [5] LLMLingua: Compressing Prompts for Accelerated Inference of Large Language Models
>
> [6] https://openai.com/blog/new-models-and-developer-products-announced-at-devday
>
> > The description of the environment setting requires further clarification. Does  \tau_int,(1)  mean that a new task will be added at every timestep? What is the maximum horizon of an episode?
>
>
> We apologize for the confusion.  \tau_int,(1) means a new task will be added after a fixed time step. The specific time value of \tau_int(1) depends on the environment complexity. We compute this value based on the minimum number of steps to complete the task in a single agent setting.
>
> Here are more details on how the task complexity and subsequently, \tau_int is determined, these can also be found in Appendix Section E.4
>
> In our approach, we initially construct a task graph delineating subgoals, which serves as the foundation for our computations. We then apply breadth-first search in a single-agent context to determine the optimal task sequence. This sequence is a key component for calculating task intervals in multi-agent collaboration scenarios. For each tool requiring activation, we incorporate its activation wait time into the respective task intervals. Additionally, for each new connection in the task graph, we increase the total task interval time by tripling the edge time. We assume each subgoal requires at least, goto, get and put 3 actions. The cumulative task interval is subsequently adjusted by a scaling factor of 0.3 and the variable $\tau$. We pick the value $tau$ ranging from 1.0, 1.5, 2.0, 2.5 and 3.0 to represent different task difficulties.  This process enables us to effectively compute task intervals tailored for multi-agent collaborative environments of varied difficulties.
>
> We have updated the manuscript to make this more clear.
>
> > The paper only highlights the emergent behavior of GPT-4 while neglecting to discuss its potential drawbacks. A more balanced perspective could be achieved by including examples of GPT-4's failure cases.
>
> We thank the reviewer for pointing this out. We have updated the paper accordingly. Here we list the common failure mode:
>
> 1.  GPT-4 heavily rely on feedback, without feedback its performance drops significantly as indicated in our Table 2.
>
> 2. GPT-4 is sensitive to prompt inputs as indicated in Table3, without inference knowledge, the performance will drop. We have updated the paper with these new discussions.

---

### Official Review · Reviewer_t1AF · 2023-11-04

**Soundness:** 3 good
**Presentation:** 4 excellent
**Contribution:** 4 excellent
**Rating:** 8
**Confidence:** 2

**Summary:**

This paper proposes a new gaming scenario and related benchmark based on a multi-agent virtual kitchen environment, CuisineWorld.  It introduces MindAgent, which demonstrates the in-context learning multiagent planning capacity of LLMs and brings several prompting techniques that help facilitate their planning ability. Extensive evaluations are conducted with multiple LLMs and prompting settings on the benchmark, including deploying the system into real-world gaming scenarios.

**Strengths:**

The work is solid and important to the community which provides a benchmark that supports the implementation of a general multi-agent infrastructure that encompasses collaboration between large language models (LLMs) and human-NPCs.

The paper is well-written and organized.

**Weaknesses:**

1.	The font size of Figure 4 is too small.
2.	The paper seems not provide a clear definition of the terms $q_{pim}$ and $c_{pim}$ in Equation 2.

**Questions:**

As weaknesses.

---

> ### Author Response · Authors · 2023-11-21
> **Response to Reviewer t1AF**
>
> We sincerely thank you for your time and constructive comments. Below, we provide detailed replies to your comments and hope we can resolve your major concerns.
>
> > The font size of Figure 4 is too small.
>
> Thanks for the suggestion, we will update the paper accordingly.
>
> > The paper seems not provide a clear definition of the terms .
>
> Thanks for pointing this out.  We apologize for the confusions. Here q refers to the utility/rewards the system will generate if the agent perform the specified action. C refers to the cost of the action. We have updated the paper to make this point more clear.

---

### Author Response · Authors · 2023-11-21
**General Response**

We thank all reviewers for their insightful comments and acknowledgment of our contributions. We highlight the major contributions of our work as follows:

* The creation and development of a new benchmark, entailing a substantial amount of effort (reviwer tSpT). Our benchmark differs from prior benchmark featuring multiple different task objectives in a single episode. Agents need to collaborate to achieve maximum collaboration efficiency.
* An infrastructure to explore the potential of LLMs in multi-agent planning (reviewer RwfK) and its potential of collaborating with human users. In addition, we perform cross-domain experiments to transfer our infrastructure into minecraft to demonstrate its generality.
* The introduction of a novel metric, CoS (Collaboration Score), to assess collaboration capabilities. (reviewer  tSpT, reviewer DUbr)
* We conduct extensive experiments involving more than two agents; we also conduct experiments with human players. (reviewer YZDa)
* We demonstrate that our framework can be transferred to a novel domain, like Minecraft. (reviewer YZDa)


We’ve revised our manuscript per the reviewers’ suggestions (highlighted in red in the uploaded revision pdf) in both main text and the appendix. Detailed responses to each reviewer’s concerns are carefully addressed point-by-point. Below summarize the major updates we’ve made:

* We updated the font size of Figure 4.
* We provide more clear definitions for the equations used in the paper.
* We updated the manuscript to reflect how we compute \tau (task interval) for different task levels.
* We add discussions on the drawbacks of leveraging LLMs as a multi-agent planner.
* We add more discussions on incorporating domain knowledge into the infrastructure.
* We add missing references and discuss their relevance to our paper.

> Gao, Yiming, et al. Towards Effective and Interpretable Human-Agent Collaboration in MOBA Games: A Communication Perspective in ICLR 2023

> Wang, Rose et al. Too many cooks: Bayesian inference for coordinating multi-agent collaboration in CogSci 2020

> Zhang, Hongxin et al. Building cooperative embodied agents modularly with large language models

> Jain, Unnat et al. A Cordial Sync: Going Beyond Marginal Policies for Multi-Agent Embodied Tasks in ECCV 2020

* We add an analysis section on the common failure modes.
* We include more details on how to transfer MineAgent to real time games.
* We add details about prompt engineering process.
* We add more details on how to transfer our infrastructure to a different gaming domain.
* We append a full interaction log to the end of the appendix.